# NUBench: A Benchmark for LLMs' Sentence-Level Negation Understanding

## Abstract

Negation is a fundamental linguistic phenomenon that poses ongoing challenges for Large Language Models (LLMs), particularly in tasks requiring deep semantic understanding. Current benchmarks often treat negation as a minor detail within broader tasks, such as natural language inference. Consequently, there is a lack of benchmarks specifically designed to evaluate comprehension of negation. In this work, we introduce *NUBench* — a novel benchmark explicitly created to assess sentence-level understanding of negation in LLMs. NUBench goes beyond merely identifying surface-level cues by contrasting standard negation with structurally diverse alternatives, such as local negation, contradiction, and paraphrase. This benchmark includes manually curated sentence-negation pairs and a multiple-choice dataset, allowing for a comprehensive evaluation of models' understanding of negation.

## 1 Introduction

Negation is a fundamental and universal phenomenon found in languages worldwide. It is closely associated with various human communicative abilities, such as denial, contradiction, deception, misrepresentation, and irony. Although affirmative statements are more common, negation still plays a significant role in language; approximately 25% of sentences in English texts contain some form of negation (Sarabi & Blanco, 2016; Hossain et al., 2020; Horn & Wansing, 2025). This prevalence and its impact on meaning make accurate interpretation of negation crucial for several natural language processing (NLP) tasks, including sentiment analysis, question answering, knowledge base completion, and natural language inference (NLI) (Khandelwal & Sawant, 2020; Hosseini et al., 2021; Singh et al., 2023). Recent studies have shown that effectively managing negation is important even for multimodal language models (Quantmeyer et al., 2024; Alhamoud et al., 2025; Park et al., 2025).

Meanwhile, negation poses significant challenges for both humans and language models. Research shows that people often find it more difficult to process and comprehend negated statements compared to affirmative ones (Wales & Grieve, 1969; Sarabi & Blanco, 2016). Similarly, many studies indicate that pretrained language models (PLMs) struggle to interpret negation accurately. For example, models like BERT (Devlin et al., 2019) and even large language models (LLMs) such as GPT-3 (Brown et al., 2020) often have difficulty distinguishing between negated and affirmative statements. These models tend to rely on superficial cues, which can result in incorrect outputs when negation is involved (Kassner & Schütze, 2020; Hossain et al., 2022a; Truong et al., 2023).

Despite its significance, there is a notable lack of dedicated evaluation benchmarks for understanding negation. Most existing resources either treat negation as a minor aspect of broader tasks or focus solely on narrow syntactic detection, often emphasizing encoder-based models (Hossain et al., 2020; Geiger et al., 2020; Truong et al., 2022; Anschütz et al., 2023). To address this gap, we introduce *NUBench* (*N*egation *U*nderstanding *Bench*mark), a dataset explicitly designed to evaluate LLMs' sentence-level comprehension of negation. Our benchmark is structured as a multiple-choice question (MCQ) task: given an original sentence, the model must select the correct standard negation from four options. The other three choices—local negation, contradiction, and paraphrase—are carefully designed distractors that test whether models truly grasp semantic scope and logical oppositions.

The contributions of this paper are summarized as follows:

- We define standard negation within the framework of sentential logic, moving beyond the cue-based and often ambiguous accounts of prior work. Grounding standard negation in logical structure not only clarifies its role in natural language but also supports the evaluation and enhancement of reasoning in LLMs.
- We create a manually curated benchmark that includes a dataset of sentence-negation pairs for fine-tuning, along with a multiple-choice evaluation task.
- We conduct systematic evaluations of decoder-based LLMs, assessing their performance under both prompting and supervised fine-tuning. This includes error and confusion analyses that highlight the models' ongoing challenges with negation.

NUBench provides valuable insights into the semantic reasoning abilities of language models and serves as a robust standard for future research focused on understanding negation.

## 2 RELATED WORK

**Negation detection and scope resolution.** Early work in negation detection and scope resolution primarily relied on rule-based systems and handcrafted heuristics, especially in domain-specific contexts like clinical texts. While these systems are effective, they lack flexibility across different domains (Chapman et al., 2001; de Albornoz et al., 2012; Ballesteros et al., 2012; Basile et al., 2012). Traditional machine learning methods, such as Support Vector Machines (SVMs) (Hearst et al., 1998) and Conditional Random Fields (CRFs) (Sutton et al., 2012), were introduced later; however, they too are limited to narrow domains (Morante et al., 2008; Morante & Daelemans, 2009; Read et al., 2012; Li & Lu, 2018).

More recently, deep learning approaches employing Convolutional Neural Networks (CNNs) (O'shea & Nash, 2015) and Bidirectional Long Short-Term Memory (BiLSTM) networks (Siami-Namini et al., 2019) have enhanced performance by providing improved contextual embeddings and sequence modeling (Fancellu et al., 2016; Bhatia et al., 2019). Pretrained transformer models like BERT have been employed through transfer learning techniques (e.g., NegBERT (Khandelwal & Sawant, 2020)), significantly increasing the accuracy of negation detection tasks. Nonetheless, these methods still largely focus on syntactic span detection, leaving deeper semantic understanding of negation a challenging area to tackle.

**Negation-sensitive subtasks of NLU.** Negation understanding has become increasingly important in natural language understanding (NLU) tasks (Hosseini et al., 2021). However, existing NLU benchmarks, such as SNLI (Bowman et al., 2015) for natural language inference (NLI), CommonsenseQA (Talmor et al., 2019) for Question Answering (QA), SST-2 (Socher et al., 2013) for sentiment analysis, STS-B (Cer et al., 2017) for textual similarity and paraphrasing, have been criticized for not adequately addressing the semantic impact of negation (Hossain et al., 2022a; Rezaei & Blanco, 2024). These datasets contain relatively few instances of negation or include negations that are not crucial to task performance, allowing language models to achieve high accuracy even when they completely ignore negation.

Recent studies, including NegNLI (Hossain et al., 2020), MoNLI (Geiger et al., 2020), and NaN-NLI (Truong et al., 2022), have introduced benchmarks for NLU that are sensitive to negation.

---



**Negate** the sentence.

Sentence: Batts are commonly used in the walls and ceilings of timber-frame buildings, rolls can be cut to size for lofts, and ropes can be used between the logs in log homes.

A (standard negation). Batts aren't typically used in the walls and ceilings of timber-frame buildings, rolls cannot be cut to size for lofts, or ropes cannot be utilized between the logs in log homes.

B (local negation). Batts are normally utilized in the walls and ceilings of timber-frame buildings, and rolls can be cut to size for lofts, but ropes cannot be utilized between the logs in log homes.

C (contradiction). Batts are rarely found in the walls and ceilings of timber-frame buildings, rolls are difficult to cut to size for lofts, and ropes are avoided between the logs in log homes.

D (paraphrase). In timber-frame buildings, batts are frequently installed in walls and ceilings, rolls can be trimmed to fit loft spaces, and ropes can be applied between the logs in log homes.

Answer: A



Figure 1: An example of NUBench multiple-choice evaluation task, where the underlined text indicates the main verb phrase of each sentence, and the red text marks the negated part.

These studies show that model performance significantly declines when negation plays a crucial role in affecting the outcome (Naik et al., 2018; Yanaka et al., 2019; Hartmann et al., 2021; Hossain et al., 2022b; Hossain & Blanco, 2022; She et al., 2023; Anschütz et al., 2023). These findings suggest that current language models tend to depend on superficial linguistic patterns rather than a genuine understanding of semantics.

**Limitations of distributional semantics.** Distributional semantics (Harris, 1954; Sahlgren, 2008) aims to create models that learn semantic representations based on patterns of word co-occurrences (Boleda, 2020; Lenci et al., 2022) and capture broad semantic relationships; however, it encounters significant challenges with negation. Negated expressions, such as "not good," often appear in similar contexts as their affirmative counterparts, like "good." As a result, models tend to generate similar vector representations for these expressions, despite their opposing meanings. Previous research has pointed out this limitation, showing that PLMs struggle to capture the subtle semantic differences introduced by antonyms and the reversal of polarity (Rimell et al., 2017; Jumelet & Hupkes, 2018; Niwa et al., 2021; Jang et al., 2022; Vahtola et al., 2022). Studies have further suggested that models like BERT find it difficult to distinguish between affirmative and negated contexts (Kassner & Schütze, 2020; Ettinger, 2020).

**Negations in generative language models.** Recent research on understanding negation has primarily focused on bidirectional models, such as BERT (Devlin et al., 2019) and RoBERTa (Liu et al., 2019), which have demonstrated strong performance in NLU and negation detection tasks. However, with the emergence of generative foundation models like GPT (Radford et al., 2018) and LLaMA (Touvron et al., 2023), attention has shifted towards evaluating how these models handle negation. Studies have shown that these generative models often exhibit a positive bias and struggle with producing or interpreting negated statements (Truong et al., 2023; Chen et al., 2023; García-Ferrero et al., 2023). Although some benchmarks, such as CONDAQA (Ravichander et al., 2022) and ScoNe (She et al., 2023), reveal these limitations, there is still a lack of robust evaluation resources specifically designed for generative models.

Building on previous studies, this paper assesses whether generative models can comprehend negation in complex sentences and identify semantic differences that extend beyond surface-level patterns.

## 3 Scope and Categorization of Negation

In this work, we aim to clarify the concept of negation by introducing a typology that clearly outlines its semantic boundaries and differentiates it from related, yet distinct, phenomena. This typology organizes various forms of meaning reversal into logically consistent categories, allowing for a more precise and systematic evaluation of how language models handle negation.

### 3.1 Typology of Negation

Negation is a fundamental semantic and syntactic operation found in natural languages, used to convey denial, rejection, or the absence of a proposition. Hereafter, we denote our negation operation for a sentence $S$ as $\text{Neg}(S)$. In formal logic, negation flips the truth value of a proposition $P$: if $P$ is true, then $\text{Neg}(P)$ is false, and vice versa. Semantically, negation creates a binary opposition between a proposition and its affirmative counterpart, meaning that each one is the opposite of the other (Horn & Wansing, 2025).

Negation can be categorized along several dimensions: scope, form, and target (see Table 1). In terms of scope, negation may affect the entire clause (referred to as *clausal negation*) or only part of it (known as *subclausal negation*). Regarding form, negation can manifest as bound morphemes, such as prefixes and suffixes (*morphological negation*), or as separate syntactic elements like "not" or "never" (*syntactic negation*). Finally, depending on its target, negation can apply to the verb (*verbal negation*) or to other elements in the sentence (*non-verbal negation*) (Zanuttini, 2001; Miestamo, 2007; Truong et al., 2022; Kletz et al., 2023).

### 3.2 Negation and Contradiction

Negation and contradiction are closely related concepts that are often conflated in NLP research (Jiang et al., 2021). Contradiction refers to the incompatibility of two propositions, meaning that they cannot both be true at the same time. While negation frequently serves as a primary mech-

Table 1: Typology of negation.

| Dimension | Negation Type | Definition | Example |
|---|---|---|---|
| Scope | **Clausal Negation ( = Sentential Negation)** | Negation that applies to the entire clause or sentence. This typically involves the use of "not", or its contracted form "n't" with auxiliary verbs. | He **speaks** English fluently. → He **doesn't speak** English fluently. |
| | **Subclausal Negation ( = Constituent / Local Negation)** | Negation that focuses on negating a specific part of a clause, such as a word or phrase, rather than the entire clause. | He speaks English **fluently**. → He speaks English, **but not fluently**. |
| Form | **Morphological Negation** | Negation expressed through affixes attached to words such as prefixes like "un-", "in-", "dis-", or suffixes like "-less". | She is **happy**. → She is **unhappy**. |
| | **Syntactic Negation** | Negation expressed through separate words (particles) in the syntax, such as "not", "never", "no", etc. | She is happy. → She is not happy. |
| Target | **Verbal Negation** | Negation that applies directly to the verb or verb phrase. | They **have finished** the work. → They **have not finished** the work. |
| | **Non-verbal Negation** | Negation that negates elements other than the verb. | There is **milk** in the fridge. → There is **no milk** in the fridge. |

Table 2: Standard negation. $P$ and $Q$ stand for propositions, respectively. In addition to *and*, *or*, and *if*, other natural language connectives such as *when* are also considered, and their negations follow the same principles presented here depending on their function.

| Type | Definition | |
|---|---|---|
| **Base case** | If $P$ is an atomic proposition, $\text{Neg}(P)$ is the proposition where the main predicate of $P$ is negated. | |
| **Inductive step** | **Conjunction** | $\text{Neg}(P, \text{ and } Q) \equiv \text{Neg}(P), \text{ or } \text{Neg}(Q)$ |
| | | $\text{Neg}(P, \text{ but } Q) \equiv \text{Neg}(P), \text{ or } \text{Neg}(Q)$ |
| | **Disjunction** | $\text{Neg}(P, \text{ or } Q) \equiv \text{Neg}(P), \text{ and } \text{Neg}(Q)$ |
| | **Implication** | $\text{Neg}(\text{if } P, Q) \equiv \text{Neg}(\text{Neg }(P), \text{ or } Q) \equiv P, \text{ and } \text{Neg}(Q)$ |
| | | $\text{Neg}(P \text{ if and only if } Q) \equiv \text{Neg}(\text{if } P, Q, \text{ and if } Q, P)$ |

anism for creating contradictions—by reversing the truth value of a proposition—contradictions can also arise from antonymy, numeric mismatches, or differences in structure and lexicon (further details can be found in Appendix A). For instance, the statements "An individual was born in France" and "An individual was born in Italy" are contradictory, but they are not negations, as the second statement does not reverse the truth of the first.

Many previous studies have overlooked the possibility that contradictions can exist independently of explicit negation. Recognizing this gap, we specifically examine the ability of LLMs to differentiate between negations and non-negated contradictions, highlighting the nuanced semantic distinctions that are involved.

## 3.3 STANDARD NEGATION

*Standard negation* refers to the typical form of negation applied to the declarative verbal main clause. It specifically negates the verb in a *main clause* (Miestamo, 2000). A main clause can function as a complete sentence on its own, consisting at minimum of a subject and a predicate. This definition is grounded in the notion that the verb acts as the head of the clause (Miller & Miller, 2011).

Building on this traditional understanding, we treat standard negation as *the process of reversing the truth value of the verb phrase in the main clause*, which we will refer to as the *main predicate* in this paper. A verb phrase is headed by a verb and can consist of a single verb or a combination of auxiliaries, complements, and modifiers (e.g., "will call" and "is being promoted") (Lakoff, 1966). Since the main predicate conveys the core action or state of the clause, negating it effectively reverses the proposition of the entire sentence. In this context, we treat standard negation as a *truth-functional operation that maps the main predicate to its complement set within the semantic space*.

We further clarify the scope of standard negation within the typology presented in Table 1. Standard negation includes both *clausal negation* and *verbal negation*, as it reverses the meaning of the entire sentence by negating the main predicate. In terms of form, standard negation can employ both *syntactic* and *morphological negation*.

Table 3: Typology of local negation.

| Type | Structure Explanation | Local Negation Example |
|------|----------------------|------------------------|
| **Relative clause negation** | A relative clause is a type of dependent clause that gives extra details about a noun or noun phrase in the main sentence. It usually begins with a relative pronoun such as *who, which, that, whom,* or *whose*. | The man who **owns** the car is my neighbor. → The man who **does not own** the car is my neighbor. |
| **Participle clause negation** | A participle clause is a type of dependent clause that begins with a participle (a verb form ending in *-ing* or a past participle). It acts like an adverb, giving extra details about the main clause, often showing time, reason, result, or sequence of actions. | **Walking** through the park, she found a lost wallet. → **Not walking** through the park, she found a lost wallet. |
| **Adverbial clause negation** | An adverbial clause is a dependent clause that acts like an adverb, modifying a verb, adjective, or adverb. It gives information such as time, reason, condition, or contrast. These clauses are introduced by subordinating conjunctions like *because, although,* or *while*. | She stayed inside because it **was raining**. → She stayed inside because it **was not raining**. |
| **Compound sentence with local negation** | A compound sentence consists of two or more main clauses joined by coordinating conjunctions such as *and, but,* or *or*. If only one of these clauses is negated, the negation applies only locally to that clause. | He submitted the report and **attended** the meeting. → He submitted the report and **did not attend** the meeting. |

Syntactically, standard negation often uses explicit negation particles, such as "not." Morphologically, it can involve *complementary antonyms* (for example, "alive" vs. "dead" or "true" vs. "false"), which occupy mutually exclusive semantic spaces, thus reversing the truth value of the proposition.

In contrast, other types of antonyms, such as *gradable antonyms* (e.g., "happy" vs. "unhappy") and *relational antonyms* (e.g., "buy" vs. "sell") (Lehrer & Lehrer, 1982), do not strictly reverse truth values. Therefore, they are classified as contradictions rather than standard negation in this paper.

**Atomic propositions.** While this characterization effectively defines standard negation for atomic propositions (elementary sentences that cannot be further decomposed) (Davis & Gillon, 2004), its application to complex sentences with multiple clauses requires a more thorough approach. In this paper, we treat *an atomic proposition as a sentence that contains a single main predicate*.

**Complex propositions.** Specifically, for propositions composed of multiple logically connected atomic statements, the method for reversing the truth value of the entire complex proposition can be ambiguous. In natural language, such logical structures typically appear as coordinated clauses (e.g., "$P$ and $Q$ or $R$") or comma-separated lists connected by "and" or "or" (e.g., "$P, Q,$ and $R$"). We treat these as equivalent to a sequence of binary conjunctions or disjunctions.

**Definition of standard negation.** In this paper, standard negation refers to natural-language sentential negation, which is formally treated as logical negation within the framework of sentential logic (Enderton, 2001). To address the complexities involved, we define standard negation recursively by applying it pairwise over the logical structure of a sentence until only atomic propositions remain, ensuring that the truth value of the entire sentence is reversed even when it contains multiple coordinated clauses. Our definition of Neg($\cdot$) is presented in Table 2. Conditionals of the form "if $P$, $Q$" are equivalent to "Neg($P$) or $Q$" in logic, and we adhere to this equivalence when defining their negation (more details can be found in Appendix C).

Our definition of standard negation is inspired by the framework of sentential logic (Enderton, 2001). However, it should be viewed as an operational definition rather than a strict mathematical formulation. Natural language sentences often lack explicit structural markers, such as parentheses, which are vital in well-formed logical formulas. Furthermore, coordination can appear with or without commas. Unlike formal languages, natural languages do not adhere to strict formation rules, making it challenging to map their structures unambiguously to logical formulas. Consequently, our definition cannot perfectly align with the forms used in sentential logic. Nevertheless, it provides a clear operational account of standard negation in natural languages.

## 3.4 LOCAL NEGATION

We define *local negation* as a form of negation that specifically targets a verb phrase outside the main clause. While the term is often used interchangeably with subclausal negation, our focus is solely on local negation relating to subclausal and verbal negation. This concept applies to four

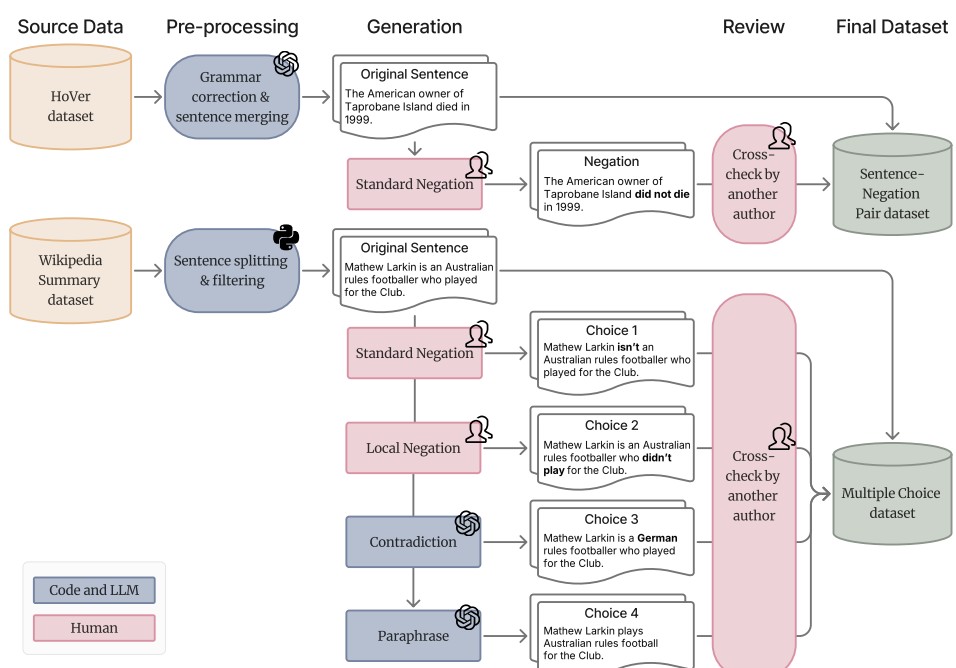

Figure 2: Dataset generation process.

types of sentence structures: relative clauses, participle clauses, adverbial clauses, and compound sentences (refer to Table 3 for more details).

In particular, conditional clauses, such as the "if $P$" part in "if $P$, $Q$" are categorized as adverbial clauses. In compound sentences, standard negation requires all main clauses to be negated in order to achieve sentence-level negation. If only a subset of the clauses is negated, this is considered local negation.

Local negation, in terms of structure, resembles standard negation, typically using explicit negation markers like "not." However, its scope is confined to a specific part of the sentence rather than encompassing the entire main clause. Because explicit cues such as "not" are still present, models that depend on shallow cue detection may be misled, failing to distinguish between standard negation and local negation.

## 4 NUBench Dataset

We construct the NUBench dataset through three main stages: (1) pre-processing, (2) generation, and (3) review. The overall workflow is illustrated in Figure 2.

**Pre-processing.** We begin by extracting sentences from two primary corpora: (1) the Hover dataset (Jiang et al., 2020), designed for multi-hop fact extraction and claim verification, and (2) the Wikipedia Summary dataset (Scheepers, 2017), which contains concise summaries from English Wikipedia. We chose these datasets because their factual content and complex sentence structures are well-suited for developing a dataset aimed at understanding standard negation in complex, sufficiently lengthy sentences. Additionally, we automatically correct any grammatical errors and merge or split sentences as needed to create well-formed single-sentence units.

**Generation.** We create two types of datasets from the pre-processed sentences: the *sentence-negation pair dataset* and the *multiple choice dataset*. In the sentence-negation pair dataset, each original sentence is paired with a manually crafted standard negation, as detailed in Section 3.3. In the multiple-choice dataset, each original sentence is presented with four options: a standard negation, a local negation, a contradiction, and a paraphrase. Each of them are described in Table 4. Together, these categories assess whether models truly understand semantic negation rather than relying on superficial cues.

Standard and local negation options are manually created rather than generated by LLMs. We have observed that LLMs often struggle to produce correct standard negations, frequently resulting in

Table 4: Multiple choice categories included in NUBench.

| Category | Description |
|---|---|
| **Standard Negation** | This category involves reversing the truth value of the main clause, which is the primary focus of the benchmark. |
| **Local Negation** | In this case, negation is applied to a subordinate clause or a partial structure, which does not reverse the entire sentence. |
| **Contradiction** | This category introduces conflicts with the original meaning through semantic changes, such as the use of antonyms, different numbers, or other entities, without employing explicit negation. |
| **Paraphrase** | Here, the original meaning is preserved while the surface form is altered. Examples of paraphrases are intentionally constructed to vary the sentence structure and word choice significantly, ensuring that no additional information is added. As a result, the original sentence still entails its paraphrase. This category tests whether models mistakenly consider different surface forms as meaning reversals, even when the semantic meanings remain equivalent. |

Table 5: NUBench statistics.

| Dataset | Split | Count |
|---|---|---|
| Sentence-Negation | Train | 3,772 |
| Multiple Choice | Demonstration | 50 |
| | Test | 1,261 |
| | Total | 5,083 |

Table 6: Models used.

| Size | | Model |
|---|---|---|
| 2-3B | Pretrained | gemma-2b, Llama-3.2-3B, Qwen2.5-3B |
| | Instruction-tuned | gemma-1.1-2b-it, Llama-3.2-3B-Instruct, Qwen2.5-3B-Instruct |
| 7-8B | Pretrained | gemma-7b, Llama-3.1-8B, Mistral-7B-v0.3, Qwen2.5-7B |
| | Instruction-tuned | gemma-1.1-7b-it, Llama-3.1-8B-Instruct, Mistral-7B-Instruct-v0.3, Qwen2.5-7B-Instruct |
| API-based | | GPT-4o mini, GPT-4.1 mini, Claude Haiku 3.5 |

subclausal or local negations instead. They can also generate incorrect local negations, even when explicitly prompted to do otherwise. Since precise negation is essential to our benchmark, these options must be developed by humans to ensure the quality of the dataset. In contrast, contradiction and paraphrase options are initially created automatically using carefully designed prompts with the OpenAI API (OpenAI, 2025) and are then refined during the review process.

**Review.** All constructed data undergo a multi-stage human review process (see Appendix I). A different author, separated from the creator, cross-checks each instance, and any disagreements are addressed in regular meetings to ensure consistency. Options for contradictions are reviewed only after the corresponding standard and local negations are finalized, as they must not overlap semantically. Consequently, the earlier negations are re-examined during the contradiction review and are cross-checked by multiple authors.

The guidelines for data generation and review are continuously updated, and any previously created data are revised accordingly (see Appendix J). This protocol ensures rigorous quality control and consistency throughout the benchmark.

**Dataset statistics.** The final dataset includes a training set of sentence-negation pairs and a multiple-choice evaluation set (see Table 5). For few-shot prompting, we construct a demonstration set of 50 examples. These are carefully selected to have unique Wikipedia page indices to avoid any overlap with the test set. Furthermore, to provide the model with a balanced overview of the task, we match the distribution of local negation types (`choice2_type`) in the demonstration set to that of the overall dataset. This ensures that the demonstrations are representative and prevents the model from developing a biased strategy for specific negation types.

## 5 EXPERIMENTS

### 5.1 EVALUATION SETUP

We evaluate models under two common Multiple-Choice Question Answering (MCQA) evaluation settings: (1) a completion-based evaluation, where the model assigns probabilities to each candidate by appending it as a continuation of the prompt, and (2) an option-selection evaluation, where the model selects from labeled options (A/B/C/D). In the completion-based evaluation, we report performance using *accuracy*, while in the option-selection evaluation, we use *exact match*. To mitigate known issues, such as selection or position bias in the option-selection evaluation, we randomly

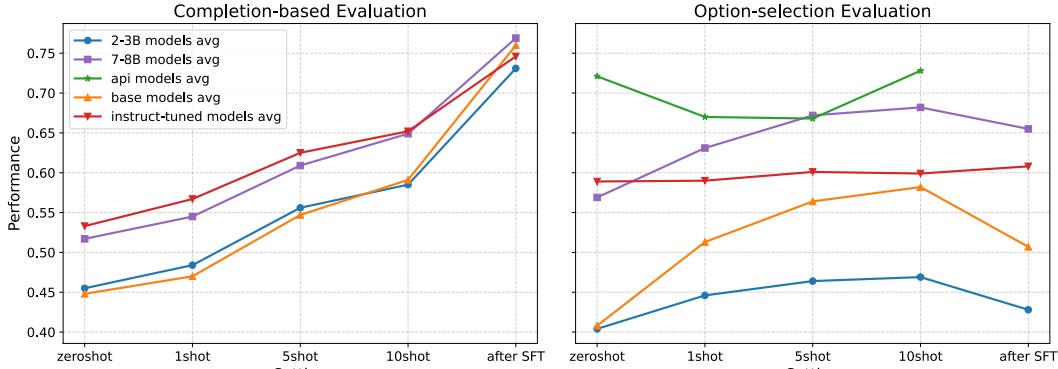

Figure 3: Model performance on NUBench. Circles (blue) represent the average performance of 2-3B models, squares (purple) indicate the average for 7-8B models, upward triangles (orange) signify the average of base models, and downward triangles (red) denote the average of instruction-tuned models. Stars (green) represent API models.

shuffle the order of the options (using random seed 42). Details of the specific prompt templates and formatting can be found in Appendix O.

**Models.** We evaluate two main groups of model sizes: those with 2-3 billion parameters and those with 7-8 billion parameters. Each group includes both pretrained models and instruction-tuned models. To also examine larger models, we incorporate API models, which are assessed only in the option-selection setting. This is because recent APIs do not provide the log-probability outputs needed for completion-based evaluation. Table 6 summarizes the models used in our experiment (Team et al., 2024; Grattafiori et al., 2024; Qwen et al., 2025; Jiang et al., 2023; Achiam et al., 2023; Hurst et al., 2024; Anthropic, 2024).

**Zero-shot and few-shot evaluation.** For each model, we evaluate performance in both zero-shot and few-shot settings using the Language Model Evaluation Harness (Gao et al., 2024). In the few-shot scenario, we use examples from the demonstration set as in-context demonstrations. Results are averaged over five random seeds (42, 1234, 3000, 5000, and 7000) and are reported for one, five, and ten examples from the demonstration set (1-shot, 5-shot, and 10-shot). We present the performance results on the test set for each model and prompt configuration.

**Supervised fine-tuning.** We conduct Supervised Fine-Tuning (SFT) using the LLaMA-Factory framework (Zheng et al., 2024) on the Sentence-Negation Pair dataset from NUBench. The dataset is formatted in the Alpaca instruction style (Taori et al., 2023). To achieve parameter-efficient training, we apply Low-Rank Adaptation (LoRA) (Hu et al., 2022) with a rank of 8, targeting all linear layers. The fine-tuning process is carried out for three epochs, using a batch size of 1, a gradient accumulation step of 8, cosine learning rate scheduling, and bfloat16 precision. After the SFT, we evaluate the model's zero-shot performance to directly assess its ability to generalize from instruction tuning without being influenced by in-context examples. It is important to note that API models are not fine-tuned, as they are not compatible with the LLaMA-Factory framework.

### 5.2 MODEL PERFORMANCE ON NUBENCH

Figure 3 displays evaluation results from all models across three settings: the zero-shot baseline, the few-shot baseline, and the zero-shot after SFT using NUBench. It summarizes the overall trends concerning model sizes, training configurations (including whether it is instruction-tuned), and evaluation settings. Complete results are reported in Appendix P.

In the completion-based evaluation, performance steadily improves from zero-shot to few-shot and further after SFT, with gaps between smaller and larger models narrowing after SFT. In the option-selection evaluation, performance also rises with more shots, but SFT yields smaller gains than few-shot prompting. Meanwhile, instruction-tuned models remain relatively stable across conditions. API models achieve the best overall performance, though their performance doesn't consistently improve with more shots, except at 10-shot.

Table 7: Error distribution and confusion analysis of pretrained and instruction-tuned Llama-3.1-8B models across various evaluation settings: zero-shot baseline, five-shot baseline, and SFT zero-shot.

| | | | | Error Rate | Incorrect Choice Distribution | | | Local Negation Confusion Rate | | | |
| --- | --- | --- | --- | --- | --- | --- | --- | --- | --- | --- | --- |
| | | | | (1-acc) | Local Negation (%) | Contra-diction (%) | Para-phrase (%) | Relative Clause (%) | Participle Clause (%) | Compound Sentence (%) | Adverbial Clause (%) |
| completion-based | Llama-3.1-8B | Baseline | zeroshot | 0.562 | 70.62 | 21.33 | 8.05 | 25.64 | 30.84 | 64.29 | 43.87 |
| | | | 5shot | 0.393 | 83.87 | 14.31 | 1.81 | 20.83 | 22.40 | 47.62 | 45.81 |
| | | SFT | zeroshot | 0.203 | 85.55 | 12.50 | 1.95 | 8.97 | 11.69 | 18.71 | 32.26 |
| | Llama-3.1-8B-Instruct | Baseline | zeroshot | 0.488 | 70.57 | 26.18 | 3.25 | 24.36 | 25.97 | 54.76 | 37.74 |
| | | | 5shot | 0.347 | 80.55 | 18.76 | 0.69 | 18.91 | 19.16 | 39.80 | 37.74 |
| | | SFT | zeroshot | 0.244 | 86.04 | 12.34 | 1.62 | 12.18 | 12.66 | 28.23 | 33.87 |
| option-selection | Llama-3.1-8B | Baseline | zeroshot | 0.541 | 47.95 | 6.74 | 45.31 | 23.08 | 24.03 | 41.50 | 19.03 |
| | | | 5shot | 0.362 | 78.95 | 6.58 | 14.47 | 30.45 | 21.75 | 40.82 | 25.16 |
| | | SFT | zeroshot | 0.444 | 58.79 | 27.34 | 13.87 | 20.83 | 20.13 | 37.41 | 20.65 |
| | Llama-3.1-8B-Instruct | Baseline | zeroshot | 0.256 | 76.53 | 15.11 | 8.36 | 16.67 | 18.83 | 19.05 | 23.23 |
| | | | 5shot | 0.269 | 74.04 | 21.83 | 4.13 | 20.83 | 16.88 | 25.17 | 19.35 |
| | | SFT | zeroshot | 0.254 | 72.32 | 19.03 | 8.65 | 15.06 | 17.86 | 21.09 | 14.52 |

## 5.3 ANALYSIS OF NEGATION UNDERSTANDING PERFORMANCE

We analyze model errors to evaluate the ability of our models to differentiate standard negation from similar semantic variants. Each type of local negation in our dataset is explicitly labeled based on its sentence structure: relative clause, participle clause, compound sentence, and adverbial clause, as defined in Table 3.

To identify which subtypes of local negation are most frequently confused with standard negation, we calculate the *confusion rate*. This is defined as the proportion of examples within each subtype where the model incorrectly selects the local negation option instead of the correct standard negation. For example, if 320 items are labeled as participle clause negation and the model incorrectly chooses the local negation option instead of the correct standard negation option in 32 of these cases, the confusion rate for participle clause negation would be 10%. Complete analysis results are provided in Appendix R.

We focus on the results of Llama-3.1-8B and its instruction-tuned version, as shown in Table 7. In the completion-based setting, error rates generally decrease from zero-shot to 5-shot and continue to improve after SFT, with most errors concentrated in local negation options. Within local negation, compound sentences exhibit the highest confusion but also show the largest relative improvement after SFT.

In the option-selection setting, the base model demonstrates unusually high errors for paraphrase options in the zero-shot scenario (nearly 40%). Although these errors decrease after SFT, they are partly offset by increases in errors for local negation and contradiction options. The instruction-tuned model consistently achieves lower overall error states and confusion rates than the base version. Compared to the completion-based setting, the option-selection setting results in lower confusion rates for adverbial clause negation.

Overall, these patterns highlight how different evaluation settings and model configurations lead to distinct types of errors, and how the addition of more examples or SFT affects error distribution.

## 6 CONCLUSION

In this work, we introduce NUBench, a benchmark designed to evaluate LLMs' sentence-level understanding of negation, going beyond surface cue detection. By distinguishing between standard negation, local negation, contradiction, and paraphrase, NUBench offers a comprehensive assessment of semantic comprehension. Our experiments demonstrate that while supervised fine-tuning and in-context learning can help reduce specific errors, these approaches still struggle to differentiate standard negation from closely related semantic variants. NUBench serves as a valuable diagnostic tool for analyzing the limitations of models' understanding of negation and stands as a robust benchmark for future research. Its design enables evaluation across diverse model families and settings, making it broadly applicable for studying semantic reasoning in LLMs.

ETHICS STATEMENT

This work does not involve the use of crowd-sourcing methods. Instead, all data included in the NUBench benchmark was carefully reviewed by the authors to ensure quality, relevance, and adherence to ethical standards. The datasets and tools used for training and evaluation are publicly available and used in compliance with their respective licenses. When leveraging OpenAI's text generation models, we take additional care to avoid generating or including any content that is harmful, biased, or violates privacy. All generated examples are manually reviewed to meet ethical and safety standards. We ensure no personally identifiable information or offensive content is present in the final dataset. The NUBench dataset will be released under the Creative Commons Attribution Non-Commercial Share Alike 4.0, ensuring transparency, reproducibility, and accessibility for future research.

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

## A  TYPOLOGY OF CONTRADICTION

Contradictions in natural language can arise in diverse ways that go beyond simple negation. Following the typology of De Marneffe et al. (2008), contradictions can be grouped into seven categories: antonymy, explicit negation, numeric mismatch, factive/modal inconsistencies, structural reversals, lexical incompatibilities, and conflicts based on world knowledge. These categories reflect the fact that contradiction covers a broader semantic scope than negation alone. Table 8 summarizes these types with definitions and examples.

Table 8: Contradiction types from De Marneffe et al. (2008). Contradiction covers a broader scope than negation.

| Contradiction Type | Definition | Example |
|---|---|---|
| Antonym | Contradiction caused by opposing meanings of aligned words. | The policy was a **success**. → The policy was a **failure**. |
| Negation | One sentence explicitly negates a statement in the other. | She **attended** the meeting. → She **did not attend** the meeting. |
| Numeric | Inconsistent numbers, dates, or quantities in related statements. | Totally, **ten** people were injured. → Totally, **five** people were injured. |
| Factive/Modal | Conflict in implied facts or modal possibilities due to verbs or auxiliaries. | He **managed to** enter the building. → He **did not enter** the building. |
| Structure | Syntactic rearrangement or argument swapping causes contradiction. | **Alice** hired **Bob**. → **Bob** hired **Alice**. |
| Lexical | Contradiction through incompatible verbs or phrases, not strictly antonyms. | The manager **praised** her performance. → The manager **expressed disappointment in** her performance. |
| World Knowledge | Contradiction relies on common-sense or background knowledge. | The Eiffel Tower is in **Paris**. → The Eiffel Tower is in **Berlin**. |

## B  COPULAR VERBS

Copular verbs, also known as linking verbs, are verbs that connect the subject of a sentence to a subject complement, which can be a noun, adjective, or other expression that describes or identifies

the subject. Unlike action verbs, copular verbs do not express actions but rather states or conditions. The most common copular verb in English is "to be" in its various forms (am, is, are, was, were). Other examples include "seem," "appear," "become," "feel," "look," "sound," "taste," and "smell" when used to describe the subject's state (Hengeveld, 1986).

As discussed in Section 3.3, standard negation in this work targets the main predicate of a clause. For sentences with copular verbs, this means that the entire verb phrase, including the copular verb and its complement, is subject to negation. For example, in the sentence "She is a doctor," the main predicate is "is a doctor." Negating this sentence results in "She is not a doctor," where the negation applies to the entire predicate, not just the verb "is."

Negation of a verb phrase including a copular verb can be realized either syntactically (e.g., "is **not** an expert") or by replacing the complement with its complementary antonym (e.g., "is a **non-expert**"), both of which result in the reversal of the main predicate's truth value. Although such constructions may superficially appear to be non-verbal negation, especially when the complement is a noun or adjective, they are, in fact, instances of verbal negation, since the negation applies to the predicate as a whole.

## C  NEGATION OF IMPLICATIONS

Negating implications presents challenges, as natural language intuitions often diverge from the rules of formal logic. Let's say there is a conditional statement, "If I study hard, I will pass the bar exam." Formally, let $P$ denote "I study hard" and $Q$ denote "I will pass the bar exam." In classical logic, the conditional "if $P$, $Q$" can be false only when $P$ is true and $Q$ is false. This implies that the negation of the conditional is "$P$ and $\text{Neg}(Q)$" ("I study hard and I won't pass the bar exam,) while the conditional itself is equivalent to "$\text{Neg}(P)$ or $Q$" ("I don't study hard or I will pass the exam) (Nguyen et al., 2023).

Psychological studies confirm that people often accept both "if $P$, $\text{Neg}(Q)$" ("If I study hard, I won't pass the exam.") and "if $\text{Neg}(P)$, $Q$" ("If I don't study hard, I will pass the exam."). However, the former can be interpreted as "$\text{Neg}(P)$ or $\text{Neg}(Q)$", and the latter "$P$ or $Q$", both of which are not equivalent to the original statement's negation, "$P$ and $\text{Neg}(Q)$". "if $\text{Neg}(P)$, $\text{Neg}(Q)$" ("If I don't study hard, I won't pass the exam.") is not the correct negation as well, as it is equivalent to "$P$ or $\text{Neg}(Q)$" (Espino & Byrne, 2012).

While humans often struggle to distinguish the correct negation of a conditional from invalid alternatives, the logical form is unambiguous. We therefore include conditional statements in our benchmark to test whether language models, like humans, are prone to intuitive but invalid interpretations, or whether they can correctly apply truth-functional reasoning.

Note that implications in natural language are not limited to the explicit "if $P$, $Q$" form, but may also appear with connectives such as *when*, *as long as*, or *unless*, which functionally convey conditional meaning and are treated under the same negation principle.

## D  COMPOUND SENTENCES AND COORDINATING CONJUNCTION

A compound sentence consists of two or more independent clauses joined by a coordinating conjunction. Each clause can stand alone, but they are combined to express related ideas (Gleitman, 1965).

Coordinating conjunctions connect elements of equal grammatical rank. The seven common ones in English are: *for, and, nor, but, or, yet, so* (often remembered as FANBOYS). Among these, *and*, *or*, and *but* are indisputably used to coordinate clauses. The others can be ambiguous or function in non-coordinating roles(e.g., indicating cause or result rather than logical structure). These are the examples using *and*, *or*, and *but* to connect sentences equally.

- "She studied hard, **and** she passed the exam."
- "I wanted to go, **but** it was raining."
- "You can call me, **or** you can send an email."

We consider only the coordinating conjunctions *and*, *or*, and *but* as indicators of compound sentences, in which two or more independent clauses are equally connected. Although *but* introduces a

contrast semantically, in terms of logical structure, it functions as a conjunction equivalent to *and*; therefore, its negation follows the same principle.

## E  LOCAL NEGATION CONSTRUCTIONS EXCLUDED

In constructing the NUBench dataset, we consider various types of local (i.e., subclausal) negation, where negation applies to a phrase or constituent rather than the main predicate. However, several constructions are excluded due to their semantic ambiguity, syntactic irregularity, or misalignment with the benchmark's focus on verbal negation.

**Infinitive Phrase Negation.**    Infinitive phrases (e.g., "to go") can be negated with "not" (e.g., "not to go" or "to not go"). Unlike the clause-level structures that define our local negation category, infinitive phrases are not full clauses but simply part of a verb phrase, making them less compatible with our definition. Moreover, although grammatically correct, this construction is relatively rare and sounds awkward depending on the context.

- **Original**: George wants to go to the park.
- **Negated (infinitive)**: George wants not to go / George wants to not go to the park.

For these reasons, we exclude infinitive phrase negation from the benchmark.

**Appositive Clause Negation.**    Appositive clauses are noun phrases that provide descriptive clarification. Attempting to negate an appositive typically involves lexical replacement rather than syntactic negation.

- **Original:** My brother, a talented musician, plays the guitar.
- **Negated (appositive):** My brother, not a talented musician, plays the guitar.

Such changes alter descriptive content rather than reversing the meaning of the predicate, and often fall into the domain of contradiction. Accordingly, they are excluded from the dataset.

**Prepositional Phrase Negation.**    Negating a prepositional phrase often involves replacing the preposition with its antonym (e.g., "with" → "without", "in" → "outside"), which results in a sentence that differs in content, rather than reversing the meaning of the predicate.

- **Original**: She went to the park with her bird.
- **Negated (preposition)**: She went to the park without her bird.

Since such modifications do not negate the verb but instead change the nature of an adjunct or argument, they fall outside the scope of standard negation or local negation in this work and are excluded.

In all of the above cases, the negation does not target the whole verb phrase but rather peripheral elements within the sentence. As the NUBench is designed to evaluate verbal negation, these local or phrase-level forms of negation were intentionally left out.

## F  DOUBLE NEGATION

Double negation refers to the use of two forms of grammatical negation within a single sentence. In standard English, only one negative form should be present in a subject-predicate construction; the presence of two negatives is generally considered non-standard and often results in an unintended meaning. For example, while "He's going nowhere" is correct, "He's not going nowhere" is ungrammatical. Another example is "I won't bake no cake," which combines verb negation ("won't") with object negation ("no cake"), resulting in a grammatically incorrect construction (Déprez et al., 2015).

In English, certain double negation constructions convey affirmative meanings rather than intensifying negation, effectively paraphrasing the original positive statement (e.g., $\neg\neg p \approx p$) (Van der Wouden, 1996). This rhetorical device, known as litotes, often manifests in expressions such as "not bad," implying "good," or "not unhappy," implying "happy." Leveraging this phenomenon, we have generated paraphrase candidates for our dataset using such double negation patterns. For example,

- **Sentence:** His characteristic style fuses samba, funk, rock and bossa nova with lyrics that blend humor and satire with often esoteric subject matter.
  **Double Negation:** His characteristic style **does not fail to fuse** samba, funk, rock, and bossa nova with lyrics that blend humor and satire with often esoteric topics.
- **Sentence:** It covers a broad range of fields, including the humanities, social sciences, exact sciences, applied sciences, and life sciences.
  **Double Negation:** It **does not exclude** a broad range of fields, including the humanities, social sciences, exact sciences, applied sciences, and life sciences.
- **Sentence:** Sanders was honoured to meet with many world dignitaries and representatives of UNESCO member nations, and delighted when delegates from UNESCO, visited Toowoomba in 2018 in return.
  **Double Negation:** Sanders **was not unhappy** to meet with many world dignitaries and representatives of UNESCO member nations, and not displeased when delegates from UNESCO visited Toowoomba in 2018 in return.

However, upon closer examination, these paraphrase candidates do not always preserve the exact meaning of the original sentence. The antonyms used (e.g., "exclude" for "cover," "unhappy" for "honoured") are not always true complementary antonyms, which does not effectively negate the meaning. Moreover, the litotes construction ("does not fail to fuse") tends to add an emphatic nuance, rather than being a perfect semantic equivalent. Therefore, the boundary between paraphrasing and double negation is ambiguous, and their relationship requires more careful analysis. Given these issues, and because our primary focus is on standard negation, we ultimately decide to exclude double negation constructions as paraphrase candidates from our dataset.

## G  HOVER DATASET

Table 9: Details of HoVer dataset structure with examples.

| Column | Detail | Example |
|---|---|---|
| id | Unique claim identifier | 0 |
| uid | User/annotator identifier | 330ca632-e83f-4011-b11b-0d0158145036 |
| claim | The statement to be verified, often requiring multi-article evidence | Skagen Painter Peder Severin Krøyer favored naturalism along with Theodor Esbern Philipsen and the artist Ossian Elgström studied with in the early 1900s. |
| supporting_facts | List of Wikipedia article titles and sentence indices providing evidence | [ { "key": "Kristian Zahrtmann", "value": 0 }, { "key": "Kristian Zahrtmann", "value": 1 }, { "key": "Peder Severin Krøyer", "value": 1 }, { "key": "Ossian Elgström", "value": 2 } ] |
| label | Whether the claim is supported | 1: SUPPORTED or 0: NOT_SUPPORTED |
| num_hops | Number of articles required for verification | 2∼4 |
| hpqa_id | Reference to the original HotpotQA pair | 5ab7a86d5542995dae37e986 |

The HoVer (**Ho**ppy **Ver**ification) dataset is developed for the tasks of multi-hop evidence retrieval and factual claim verification. In HoVer, each claim requires supporting evidence that spans multiple English Wikipedia articles to determine whether the claim is substantiated or not. The dataset is distributed under a CC BY-SA 4.0 License, and it can be accessed via its official homepage[1]. Table 9 offers an overview of the dataset's structure. The data is split into training, validation, and test sets, containing 18,171, 4,000, and 4,000 examples respectively.

HoVer is constructed on top of the HotpotQA dataset, which is designed to evaluate multi-hop reasoning in question answering. HotpotQA itself is a large-scale collection of Wikipedia-based QA pairs created to address the limitations of prior QA datasets, which often fail to require complex reasoning or explanatory answers (Yang et al., 2018). The construction of HoVer involves rewriting HotpotQA question-answer pairs into claim statements, which are then validated and labeled by annotators. Claims are extended to require multi-hop evidence from up to four Wikipedia articles and are systematically modified to increase complexity. Final labels are assigned as SUPPORTED or NOT-SUPPORTED (Jiang et al., 2020).

Table 10: Details of Wikipedia Summary dataset structure with examples.

| Column | Detail | Example |
|---|---|---|
| title | Article title from Wikipedia. | Alain Connes |
| description | A brief description or category for the article (when available). | French mathematician |
| summary | The extracted summary or introduction section of the article, typically more concise than the full text. | Alain Connes (; born 1 April 1947) is a French mathematician... |
| full_text | The complete article text (when included), encompassing the full body of the Wikipedia page. | Alain Connes (; born 1 April 1947) is a French mathematician... |
| __index_level_0__ | Index number for each entry in the dataset. | 3 |

## H  WIKIPEDIA SUMMARY DATASET

The Wikipedia Summary Dataset contains the titles and introductory summaries of English Wikipedia articles, extracted in September 2017. A summary or introduction in this context refers to the content from the article title up to the content outline (i.e., before the first section heading). The dataset was originally released via GitHub[2], but is now accessible through the Hugging Face Hub[3]. The dataset license is not explicitly mentioned, but as the original Wikipedia data is distributed under the CC BY-SA 4.0, it is assumed that the dataset would be distributed under the same license. For licensing details, refer to the Wikimedia Terms of Use [4]. Table 10 offers an overview of the dataset's structure. The dataset comprises approximately 430,000 articles, only providing the training set (Scheepers, 2017).

## I  HUMAN REVIEW PROTOCOL

To ensure high-quality data construction, we implement a rigorous quality control protocol that combines generation, independent review, and iterative consensus building. The process involves the following key steps:

- **Task allocation and independence.** Authors are assigned distinct portions of the dataset, but no author is permitted to review the data they have generated. This ensures that each instance is subject to at least one independent review.
- **Sequential authoring across choices.** For the multiple-choice dataset, construction proceeds in four stages: standard negation, local negation, contradiction, and paraphrase. At each stage, different authors are responsible for creating the new option, while reviewers who have not authored that option perform the verification.
- **Cross-checking and layered review.** Each newly created option is reviewed by at least one other author, and reviewers also revisit earlier options in the same instance. For example, when reviewing the paraphrased sentence, the reviewer also checks that standard negation, local negation, and contradiction sentences are correct. As a result, every instance undergoes multiple rounds of verification across stages, such that all authors ultimately examine data they have not created themselves.
- **Guideline refinement and retroactive correction.** Generation and reviewing guidelines are continuously updated based on discussion of ambiguous or problematic cases. Whenever the guidelines changes, all previously created data are revisited to ensure compliance, promoting consistency across the dataset.
- **Consensus and adjudication.** Disagreements are discussed in weekly meetings and, if necessary, adjudicated by a lead reviewer, ensuring that no instance remains unresolved.

Overall, this iterative and layered procedure ensures that every instance in the multiple-choice dataset is independently reviewed multiple times, leading to stable guidelines and a consistent dataset.

## J  DETAILED PRINCIPLES AND EXAMPLES OF THE NUBENCH

While the main text already defines the core notions of standard and local negation (Section 3) and explains how they are applied throughout dataset construction (Section 4), here we provide more detailed illustrations.

---

[1] https://hover-nlp.github.io/

[2] https://github.com/tscheepers/Wikipedia-Summary-Dataset

[3] https://huggingface.co/datasets/jordiclive/wikipedia-summary-dataset

[4] https://foundation.wikimedia.org/wiki/Policy:Terms_of_Use

**Paraphrasing before Negation.** Before negating, the main verb or other components may be paraphrased with synonyms, provided that the sentence's tense, structure, and meaning remain strictly equivalent before applying standard negation. Authors refer to the Merriam-Webster Thesaurus [5]. For example,

- **Original Sentence**: Toumour is a village and rural commune in Niger **located near** the Niger–Nigeria **border**.
  - **Paraphrased Sentence**: Toumour is a village and rural commune in Niger **that is found close to** the Niger–Nigeria **boundary**.
    → **Standard Negation after Paraphrase**: Toumour **isn't** a village and rural commune in Niger that is found close to the Niger–Nigeria boundary.
  - **Explanation**: In this example, the participle clause "located near the Niger–Nigeria border" is rephrased as a relative clause "that is found close to the Niger–Nigeria boundary." Since both constructions serve as modifiers and preserve the same semantic role, we treat them as equivalent in meaning for the purpose of standard negation.

- **Original Sentence**: The armed forces **said** Boko Haram **attacked** their military post on March 15, 2020, which they responded to by repelling the attack, killing 50 insurgents.
  - **Paraphrased Sentence**: The armed forces **stated** that Boko Haram **assaulted** their military post on March 15, 2020, which they responded to by repelling the attack, killing 50 insurgents.
    → **Standard Negation after Paraphrase**: The armed forces **didn't state** that Boko Haram assaulted their military post on March 15, 2020, which they responded to by repelling the attack, killing 50 insurgents.
  - **Explanation**: In this example, the reporting verb "said" is paraphrased as "stated," and the verb "attacked" is replaced with the synonym "assaulted." These substitutions preserve the original tense and meaning, allowing standard negation to be applied without altering the semantic content of the sentence.

**Negation of Simple Sentences.** For simple, declarative sentences, standard negation is achieved by inserting "not" after the auxiliary or main verb, or by replacing the predicate with its complementary antonym. For example, "She is happy." → "She is not happy."; "The room is occupied." → "The room is unoccupied."

**Negation in Compound Sentences.** When multiple clauses or propositions are coordinated (e.g., with "and", "or", "but"), standard negation is logically applied, governed by De Morgan's laws. Here, "but" is treated as a coordinating conjunction equivalent to "and" in terms of logical structure, so its negation follows the same principle.

- Conjunction "$P$ and/but $Q$": the negation is "Neg($P$) or Neg($Q$)".
- Disjunction "$P$ or $Q$": the negation is "Neg($P$) and Neg($Q$)".

For example, "He passed the test and received an award." is negated as "He did not pass the test or did not receive an award."

When application of logical negation produces unnatural language, sentences may be split or slightly rephrased for fluency, provided logical meaning is preserved. For example,

- **Original:** "He finished the report and submitted the assignment."
- **Standard Negation:** "He did not finish the report or did not submit the assignment."
- **Standard Negation, but Splitted:** "He did not finish the report. Or, he did not submit the assignment."

**Coordinated Elements in the Sentence.** When a sentence contains coordinated elements (such as subjects, objects, or predicates connected by "and" or "or"), standard negation typically follows logical principles derived from De Morgan's Laws. However, whether logical negation applies to

---

[5]https://www.merriam-webster.com/

each individual component or to the entire predicate as a whole depends on whether the coordination expresses multiple independent propositions or a single collective event.

- If the coordination introduces semantically distinct propositions, that is, each conjunct could form a complete sentence on its own, negation must be applied to each proposition individually. For example, "My sister and I studied hard."
  This sentence can be interpreted as: "My sister studied hard and I studied hard."
  Therefore, the correct standard negation is: "My sister did not study hard, or I did not study hard."

- Conversely, if the coordination connects elements that jointly participate in a single action or state (e.g., a shared subject or a collective predicate), then the sentence is treated as a simple clause, and the predicate as a whole is negated. Logical decomposition is not appropriate. For example, "My sister and I share clothes."
  This expresses a single collective action involving both participants.
  Therefore, the correct standard negation is: "My sister and I do not share clothes."
  (NOT: "My sister does not share clothes, or I do not share clothes.")

- This distinction is crucial: even if two noun phrases are coordinated, if the sentence semantically decomposes into separate atomic propositions, standard negation must apply to each atomic proposition. Otherwise, it applies to the whole predicate as one unit.

- Other examples of semantically collective predicates where logical splitting is not appropriate include: "be the same", "have in common", "do something together", "combine", "unite", etc. These describe inherently joint or relational properties, not independent propositions. For example, "Clarence Brown and Peter Glenville are from the same country." should be negated as "Clarence Brown and Peter Glenville are not from the same country."

**Use of Antonyms.** When replacing predicates with antonyms in standard negation, only complementary antonyms are appropriate, as they provide a clear binary opposition, ensuring logical consistency of negation. Gradable and relational antonyms are unsuitable for standard negation because their antonyms do not represent the logical complement of the original predicate. In other words, replacing a predicate $p$ with its antonym does not produce $\neg p$ in a truth-conditional sense.

Specifically, unlike complementary antonyms, which form mutually exclusive pairs (i.e., $p \cup \neg p = U$ and $p \cap \neg p = \emptyset$), gradable and relational antonyms do not partition the meaning space cleanly, and thus fail to reverse the truth value reliably.

- **Complementary Antonyms**: Also called binary/contradictory antonyms. These antonyms represent mutually exclusive pairs with no intermediate states. The presence of one implies the absence of the other. Examples include:
  - alive / dead
  - true / false
  - present / absent
  - occupied / vacant

  Using complementary antonyms in negation ensures a direct and unambiguous reversal of the original proposition's truth value.

- **Gradable Antonyms**: These antonyms exist on a continuum and allow for varying degrees between the two extremes. Negating one does not necessarily affirm the other. Examples include:
  - hot / cold
  - happy / sad
  - tall / short
  - young / old

  Due to their scalar nature, gradable antonyms are inappropriate for standard negation, as they do not provide a definitive binary opposition.

- **Relational Antonyms**: Also known as converse antonyms, these pairs describe a reciprocal relationship where one implies the existence of the other. Examples include:

     – parent / child

     – teacher / student

     – buy / sell

     – employer / employee

Relational antonyms are context-dependent and do not represent direct opposites in a binary sense, making them unsuitable for standard negation purposes.

**General Principles of Standard Negation.**

- The negated sentence must preserve all elements (subject, tense, objects, adjuncts, etc.) of the original, except for the truth value of the main predicate.

- When naturalness and logical negation conflict, logical correctness takes priority, but minimal rephrasing is allowed for fluency.

- If the negated clause creates a contradiction with other parts of the sentence, the contradictory clause must be removed. For example, the standard negation of the sentence "While the spatial size of the entire universe is unknown, it is possible to measure the size of the observable universe, which is approximately 93 billion light-years in diameter."
  will be "While the spatial size of the entire universe is unknown, it isn't possible to measure the size of the observable universe."
  The relative clause must be removed because its content directly contradicts the negated main clause.

**Common Negation Errors and Corrections.**

- **Original sentence**: His characteristic style fuses samba, funk, rock **and** bossa nova with lyrics that blend humor and satire with often esoteric subject matter.

  - **Incorrect negation**: His distinctive style **doesn't fuse** samba, funk, rock **or** bossa nova with lyrics that blend humor and satire with often esoteric subject matter.

  - **Correct negation**: His distinctive style **doesn't fuse** samba, funk, rock **and** bossa nova with lyrics that blend humor and satire with often esoteric subject matter.

  - **Explanation**: The verb "fuse" implies a combination of all listed elements. "and" must be preserved.

- **Original sentence**: The mascot of Avon Center School is the "Koalaty Kid," **while** the mascot at Prairieview **is** an eagle **and** the mascot at Woodview **is** an owl.

  - **Incorrect negation**: Avon Center School's mascot is not the "Koalaty Kid," Prairieview's mascot **is not** an eagle, **or** Woodview's mascot **is not** an owl.

  - **Correct negation**: Avon Center School's mascot is not the "Koalaty Kid," **while** the mascot at Prairieview **is** an eagle **and** the mascot at Woodview **is** an owl.

  - **Explanation**: Two clauses connected by while are not coordinated propositions (as with *and* or *or*), but instead express contrastive information. Therefore, applying logical negation across both clauses is incorrect. Negation should apply only to the main clause (here, the first statement), while the contrasting clause remains affirmative.

## K   CODE FOR DATA CONSTRUCTION

### K.1   SENTENCE-NEGATION PAIR DATASET

To construct the sentence-negation pair dataset, we begin by randomly sampling sentences labeled as "supported facts" from the HoVer dataset. Since the original data often contains grammatical errors, we utilize OpenAI's API (OpenAI, 2025) to automatically correct these issues. In cases where the selected text consists of multiple sentences, we merge or split them as needed to ensure that each example is a single sentence, aligning with our sentence-level task objective.

We select different model versions depending on the complexity of each task. For sentence merging, which demands nuanced contextual understanding and complex syntax, we use `GPT-4`. For grammar correction, where edits are more straightforward, `GPT-3.5` is sufficient.

```
1  def grammar_fix(claim):
2      messages = [{"role": "system", "content": "Fix grammatical errors."},
3      {"role": "user", "content": f"If there are errors, please fix the sentence: {claim} \n If
         there aren't, return the original sentence. Provide only the resulting sentence without
         any additional explanation or introduction."}]
4      response = client.chat.completions.create(model="gpt-3.5-turbo", messages=messages)
5      fixed_text = response.choices[0].message.content.strip()
6      return fixed_text
```

Listing 1: Fixing Grammar with OpenAI API.

```
1  def merge_sentences_with_gpt(claim):
2      messages = [{"role": "system", "content": "Merge sentences into a single one."},
3      {"role": "user", "content": f"Merge these sentences: {claim} \n Provide only the resulting
         sentence without any additional explanation or introduction."}]
4      response = client.chat.completions.create(model="gpt-4-turbo-preview", messages=messages)
5      merged_text = response.choices[0].message.content.strip()
6      return merged_text
```

Listing 2: Merging Sentences with OpenAI API.

### K.2 MULTIPLE CHOICE DATASET

To construct the multiple-choice dataset, we first segment the "summary" column of the Wikipedia Summary dataset, which often contains multiple sentences in a single entry, into individual sentences. To focus on the challenges of negation in complex sentences, we filter out sentences that are too short. This process is done with Python code.

Since conditional sentences (e.g., "If $P$, $Q$") are rarely present in the Wikipedia summary dataset, we adopt a two-step approach: (1) prompting the model to generate conditional variants from given sentences (using OpenAI API, GPT-4o-mini), and (2) manually filtering or lightly editing the results to obtain valid conditionals.

Subsequently, we automatically generate contradictions and paraphrases for each sentence via the OpenAI API (GPT-4o) as well, followed by human review. The following scripts illustrate the procedures.

```
1  import pandas as pd
2  import re
3  from datasets import load_dataset
4  import random
5
6  df = pd.DataFrame(load_dataset("jordiclive/wikipedia-summary-dataset")['train'].shuffle(seed
       =42).select(range(10000)))
7  df = df.drop(columns=['full_text'])
8
9  def split_into_sentences(text):
10     sentences = re.split(r'(?<=[.!?]) +', text)
11     return sentences
12
13 df['sentence'] = df['summary'].apply(split_into_sentences)
14 df = df.explode('sentence')
15 df = df[df['sentence'].apply(lambda x: len(x.split()) >= 30)]
16 df = df.reset_index(drop=True)
17 df.to_csv("file/wikipedia_summary_sentences.csv", index=False)
```

Listing 3: Sentence extraction and preprocessing from Wikipedia summaries.

```
1  def generate_conditionals(sentence):
2      prompt = f"""
3      Based on the sentence below, write a conditional sentence that uses the main topic of the
        sentence.
4      The conditional sentence should express a hypothetical situation or cause-effect
        relationship related to the topic. It can be slightly complex in structure.
5      For example:
6      - If it rains tomorrow, I will stay home.
7      Sentence:
8      '{sentence}'
9      """
10
11     completion = client.chat.completions.create(
12         model="gpt-4o-mini",
13         messages=[
14             {"role": "system", "content": "You are a helpful assistant that specializes in
        generating conditional sentences."},
```

```
15          {"role": "user", "content": prompt}
16      ]
17   )
18
19   return completion.choices[0].message.content
```

Listing 4: Conditionals sentence generation.

```
 1  def generate_contradiction(sentence):
 2      prompt = f"""
 3      You will be given a sentence. Generate a contradictory sentence that directly conflicts
         with the original sentence without using standard negation.
 4
 5      Definitions:
 6      - Standard negation: Directly negating the main verb or using words like 'not', 'no', '
         never', or negative contractions such as \"isn't\", \"doesn't\", or \"can't\".
 7      - Contradiction: A sentence that logically conflicts with the original statement. The
         contradiction must be such that both sentences cannot logically be true at the same time
         under any circumstances.
 8
 9      Important:
10      - Do not change the main verb from the original sentence.
11      - Do not use 'never' or other negative words to form the contradiction.
12      - Ensure the contradicted sentence logically excludes the possibility of the original
         sentence being true simultaneously.
13
14      Examples:
15      Original sentence: \"The tallest student won the award.\"
16      Contradicted sentence: \"The shortest student won the award.\"
17
18      Original sentence: \"The room was completely dark.\"
19      Contradicted sentence: \"The room was brightly lit.\"
20
21      Original sentence: \"The event took place in the morning.\"
22      Contradicted sentence: \"The event took place in the evening.\"
23
24      Original sentence: \"All people are dying.\"
25      Contradicted sentence: \"Some people are dying.\"
26
27      Now, generate a contradictory sentence without standard negation, without changing the
         main verb, and ensuring the two sentences are logically incompatible, for the following:
28
29      Original sentence: \"{sentence}\"
30
31      Contradicted sentence:
32      """
33
34      completion = client.chat.completions.create(
35          model="gpt-4o",
36          messages=[
37              {"role": "system", "content": "You are a helpful assistant tasked with generating
         logical contradictions. Do not use negation to make contradiction."},
38              {"role": "user", "content": prompt}
39          ]
40      )
41
42      return completion.choices[0].message.content
```

Listing 5: Contradiction generation.

```
 1  def generate_paraphrase(sentence):
 2      prompt = f"""
 3          Paraphrase the following sentence using synonyms or slight structural variations
         without changing its meaning.
 4          Do not add or remove any main verbs. Keep the original intent of the sentence intact.
 5
 6          Original sentence: "{sentence}"
 7
 8          Paraphrased sentence:
 9          """
10
11      completion = client.chat.completions.create(
12          model="gpt-4o",
13          messages=[
14              {"role": "system", "content": "You are a helpful assistant skilled at generating
         paraphrases while keeping the meaning of sentences unchanged."},
15              {
16                  "role": "user",
17                  "content": prompt
18              }
```

```
19        ]
20    )
21
22    return completion.choices[0].message.content
```

Listing 6: Paraphrase generation.

## L    NUBENCH DATASET STRUCTURE

NUBench consists of two subsets: a sentence-negation pair dataset for supervised fine-tuning and a multiple-choice dataset for evaluation. Both datasets are built on English text and reviewed by authors following strict guidelines.

### L.1    SENTENCE-NEGATION PAIR DATASET

This subset contains pairs of affirmative and corresponding standard negation sentences. It includes the following fields:

- index: the index of the data.
- premise: the original sentence.
- hypothesis: its logically negated form.

### L.2    MULTIPLE-CHOICE DATASET

This evaluation set presents each original sentence with four candidate transformations.

- wikipedia_index: the original index of the Wikipedia Summary dataset.
- index: the index of the data.
- sentence: the original sentence.
- choice1: standard negation (correct answer).
- choice2: local negation (subclausal negation).
- choice2_type: specifies the type of local negation.
- choice2_element: a short description of the phrase or clause that was negated (e.g., "being built", "which crashed").
- choice3: contradiction (non-negated, semantically incompatible).
- choice4: paraphrase (semantically equivalent).

Table 11: Choice 2 Types and Distributions.

| choice2_type | Definition | Demonstration Set | Test Set |
|---|---|---|---|
| **relative_part** | negation inside relative clauses (e.g., "who did not attend...."). | 12 | 312 |
| **pp_part** | negation in participle clauses (e.g., "not walking through the park...."). | 12 | 308 |
| **adverb_part** | negation in adverbial clauses (e.g., "because it was not raining"). | 12 | 310 |
| **compound_part** | negation applied to one clause within a compound sentence. | 12 | 294 |
| **non-applicable** | used when the sentence structure does not support a valid local negation variant under our definition. | 2 | 37 |
| **Total** | | 50 | 1,261 |

The details of choice2_type and distribution on demonstration and test sets are described in Table 11. It follows the definition in Table 3.

In addition to the Wikipedia Summary dataset, we supplement the evaluation set with conditionals (e.g., If $P$, $Q$) by manually searching Wikipedia articles where such constructions are more likely to occur (e.g., Newton's laws of motion). Among the 100 conditional sentences included across demonstration and test sets, 20 are collected through manual search (marked with indices beginning with "S" in wikipedia_index). Meanwhile, the remaining 80 are sampled from the Wikipedia Summary dataset and converted into conditional form using the script in Listing 4 (Appendix K).

# M  INCORPORATING NUBENCH INTO LM EVALUATION HARNESS

This section describes how NUBench is integrated into the LM Evaluation Harness (Gao et al., 2024) for zero-shot and few-shot evaluation, both in completion-based and option-selection settings.

```
1  task: nubench_completion
2  dataset_path: {dataset_path}/NUBench
3  dataset_name: multiple-choice
4  output_type: multiple_choice
5  test_split: test
6  fewshot_split: demonstration
7  process_docs: !function utils.process_docs_completion
8  doc_to_text: "{{query}}"
9  doc_to_target: "{{gold}}"
10 doc_to_choice: "choices"
11 metric_list:
12   - metric: acc
13     aggregation: mean
14     higher_is_better: true
```

Listing 7: NUBench/NUBench_completion.yaml

```
1  task: nubench_option
2  dataset_path: {dataset_path}/NUBench
3  dataset_name: multiple-choice
4  output_type: generate_until
5  test_split: test
6  fewshot_split: demonstration
7  process_docs: !function utils.process_docs_option
8  doc_to_text: "{{query}}"
9  doc_to_target: "{{answerLetter}}"
10 generation_kwargs:
11   until:
12     - ""
13     - "\n"
14 metric_list:
15   - metric: exact_match
16     aggregation: mean
17     higher_is_better: true
18     ignore_punctuation: true
19     ignore_case: true
20 filter_list:
21   - name: get_response
22     filter:
23       - function: "regex"
24         regex_pattern: "^(.*?)(?=\\n|$)"
25       - function: remove_whitespace
26       - function: "regex"
27         regex_pattern: "^(.*?)\\s*$"
28       - function: take_first
29 dataset_kwargs:
30   trust_remote_code: true
```

Listing 8: NUBench/NUBench_option.yaml

```
1  import re
2  import datasets
3  import random
4
5  def process_docs_completion(dataset: datasets.Dataset) -> datasets.Dataset:
6      def _process_doc(doc):
7          prompt = f"Negate the sentence.\nSentence: {doc['sentence']}\nNegation:"
8
9          if doc.get("choice2_type", "") == "non-applicable":
10             choices = [doc["choice1"], doc["choice3"], doc["choice4"]]
11         else:
12             choices = [doc["choice1"], doc["choice2"], doc["choice3"], doc["choice4"]]
13
14         return {
15             "query": prompt,
16             "choices": choices,
17             "gold": 0
18         }
19     return dataset.map(_process_doc)
20
21
22 def process_docs_option(dataset: datasets.Dataset, seed: int = 42) -> datasets.Dataset:
23     rng = random.Random(seed)
24
```

```
25    def _process_doc(doc):
26        initial_prompt = f"Negate the sentence.\nSentence: {doc['sentence']}\n"
27
28        c1 = doc.get("choice1", "")
29        c2 = doc.get("choice2", "")
30        c3 = doc.get("choice3", "")
31        c4 = doc.get("choice4", "")
32        c2_na = str(doc.get("choice2_type", "")).lower() == "non-applicable"
33
34        gathered = []
35        keys = []
36
37        if c1:
38            gathered.append(c1); keys.append("choice1")
39        if c2 and not c2_na:
40            gathered.append(c2); keys.append("choice2")
41        if c3:
42            gathered.append(c3); keys.append("choice3")
43        if c4:
44            gathered.append(c4); keys.append("choice4")
45
46        assert len(gathered) >= 3, "Need at least 3 choices for multiple-choice."
47
48        paired = list(zip(gathered, keys))
49        rng.shuffle(paired)
50        choices, choice_keys = zip(*paired)
51
52        try:
53            answer_idx = choice_keys.index("choice1")
54        except ValueError:
55            answer_idx = 0
56
57        label_map = ["A", "B", "C", "D"]
58        labels = label_map[:len(choices)]
59
60        lines = [
61            "Given the following instruction and candidate answers, choose the single best
     answer.",
62            f"Instruction: {initial_prompt}",
63        ]
64        for lab, ch in zip(labels, choices):
65            lines.append(f"{lab}. {ch}")
66        joined = ", ".join(labels)
67        lines += ["", f"Your response should be one of {joined}.", "Only output the letter.",
     "Answer:"]
68        prompt = "\n".join(lines)
69
70        return {
71            "query": prompt,
72            "choices": list(choices),
73            "choice_keys": list(choice_keys),
74            "answerKey": answer_idx,
75            "answerLetter": labels[answer_idx],
76        }
77
78    return dataset.map(_process_doc)
```

Listing 9: NUBench/utils.py

## N    FINETUNING VIA LLaMA-FACTORY

We detail our supervised fine-tuning setup using LLaMA-Factory (Zheng et al., 2024) with LoRA (Hu et al., 2022) on NUBench training data, including configuration of the fine-tuning and instruction-based examples in Alpaca format (Taori et al., 2023).

The YAML configuration provided in Listing 10 is specific to the LLaMA-3.1-8B model. Other models (e.g., Qwen or Mistral) can be fine-tuned similarly by modifying the model_name_or_path and template fields in the configuration file accordingly.

```
1  ### model
2  model_name_or_path: Llama-3.1-8B
3  trust_remote_code: true
4
5  ### method
6  stage: sft
7  do_train: true
8  finetuning_type: lora
```

```yaml
 9  lora_rank: 8
10  lora_target: all
11
12  ### dataset
13  dataset: nubench_train
14  template: llama3
15  cutoff_len: 512
16  max_samples: 5000
17  overwrite_cache: true
18  preprocessing_num_workers: 16
19  dataloader_num_workers: 4
20
21  ### output
22  output_dir: lora/sft/Llama-3.1-8B
23  logging_steps: 10
24  save_steps: 500
25  plot_loss: true
26  overwrite_output_dir: true
27  save_only_model: false
28  report_to: none  # choices: [none, wandb, tensorboard, swanlab, mlflow]
29
30  ### train
31  per_device_train_batch_size: 1
32  gradient_accumulation_steps: 8
33  learning_rate: 1.0e-4
34  num_train_epochs: 3.0
35  lr_scheduler_type: cosine
36  warmup_ratio: 0.1
37  bf16: true
38  ddp_timeout: 180000000
39  resume_from_checkpoint: null
```

Listing 10: Llama-3.1-8B_lora_sft.yaml

```json
 1  [
 2    {
 3      "instruction": "Negate the sentence.",
 4      "input": "Sentence: Eddie Vedder was born before Nam Woo-hyun.",
 5      "output": "Eddie Vedder wasn't born before Nam Woo-hyun."
 6    },
 7    {
 8      "instruction": "Negate the sentence.",
 9      "input": "Sentence: Halestorm is from Pennsylvania, while Say Anything is from California.
        ",
10      "output": "Halestorm is not from Pennsylvania, while Say Anything is from California."
11    },
12    (...)
13  ]
```

Listing 11: Sentence-Negation Pair dataset for training in alpaca format.

## O  PROMPT SELECTION FOR IN-CONTEXT LEARNING

We explore a range of prompt types for in-context learning, from minimal instructions to more detailed variants (Zhao et al., 2021; Li, 2023; Wan et al., 2023). Specifically, we evaluate three prompt styles:

1. **Simple prompt**: a minimal instruction, "Negate the sentence."

2. **Definition prompt**: a concise description of the task, "Negate the main predicate of the main clause so that the proposition is logically reversed."

3. **Detail prompt**: an extended instruction with explicit step-by-step guidelines on identifying the main predicate, preserving other sentence elements, handling antonyms, and applying negation consistently across logical operators (see Listing 12 for the full format).

```
"Reverse the truth value of the main predicate (verbal phrase) in the main
    clause, while preserving all other elements of the main clause
    unchanged.

1) Identify the main clause and its main verb (main predicate). Ignore
    subordinate clauses.
2) Preserve all other main-clause content.
3) Insert a negative particle such as "not" into the main verb, or replace
    it with a complementary antonym only if it forms an absolute binary (e
    .g., alive/dead, true/false, possible/impossible).
```

```
4) If the sentence contains multiple propositions connected by logical
    operators (e.g., and, or, conditional constructions), negate it in a
    way that reverses the entire proposition (e.g., A and B => not A or
    not B; If A then B => A and not B).

Sentence: {doc['sentence']}"
```

Listing 12: Detail prompt format.

Across both completion-based and option-selection evaluation settings, we found that the simple prompt consistently achieved the highest average performance in zero-shot and few-shot settings (See results in Appendix P.) While the definition-based and detail prompts occasionally provided more explicit guidance, they did not improve performance overall. Therefore, we adopt the simple prompt as our default setting in the main experiments. The complete prompt formats, including prompt–response structures for each evaluation setting, are provided in Listings 13 and 14, based on the simple prompt.

```
Prompt] "Negate the sentence.
Sentence: Chromosome 2 is the second-largest human chromosome, spanning
    more than 242 million base pairs and representing almost eight percent
     of the total DNA in human cells.
Negation:"

Response] "Chromosome 2 isn't the second-largest human chromosome, which
    measures more than 242 million base pairs and represents almost eight
    percent of the entire DNA in human cells."
```

Listing 13: Completion-based Format.

```
Prompt] "Given the following instruction and candidate answers, choose the
    single best answer.
Instruction: Negate the sentence.
Sentence: Chromosome 2 is the second-largest human chromosome, spanning
    more than 242 million base pairs and representing almost eight percent
     of the total DNA in human cells.

A. Chromosome 2 isn't the second-largest human chromosome, which measures
    more than 242 million base pairs and represents almost eight percent
    of the entire DNA in human cells.
B. Chromosome 2 is the second-largest human chromosome, which doesn't span
    more than 242 million base pairs or represent nearly eight percent of
    the whole DNA in human cells.
C. Chromosome 2 is the smallest human chromosome, spanning fewer than 50
    million base pairs and representing less than two percent of the total
     DNA in human cells.\
D. Chromosome 2 is the second-biggest human chromosome, with over 242
    million base pairs, making up nearly 8% of all DNA in human cells.

Your response should be one of A, B, C, D.
Only output the letter.
Answer:"

Response] "A"
```

Listing 14: Option-selection Format.

## P   TOTAL MODEL PERFORMANCE ON NUBENCH

Following the classification of prompts introduced in Appendix O, we report results on NUBench separately for the simple, definition, and detail prompt styles. The experiments are conducted under three evaluation regimes: (i) zero-shot, (ii) few-shot (1-, 5-, and 10-shot, averaged across seeds), and (iii) supervised fine-tuning (SFT). For SFT, we assess models only in the zero-shot setting in order to directly measure the effect of task-specific training without the influence of in-context examples.

Table 12, Table 13, and Table 14 present the complete numerical results for the simple, definition, and detail prompts, respectively. The same results are also summarized in graphical form in Figure 3, Figure 4, and Figure 5.

Across prompt types, we observe the following trends:

- **Simple prompts.** Results for simple prompts are described in Section 5.2; we summarize them here only for completeness.

Table 12: Zero-shot, few-shot, and SFT evaluation results on NUBench with the **simple prompt.** SD denotes standard deviation across random seeds or runs. Few-shot results are averaged over five random seeds (42, 1234, 3000, 5000, and 7000) and one, five, and ten demonstration examples (1-, 5-, 10-shot). Red text indicates the model with the highest performance in each column (excluding API models).

| evaluation setting | zero-shot | | 1-shot | | 5-shot | | 10-shot | | After SFT | |
|---|---|---|---|---|---|---|---|---|---|---|
| | comple-tion | option | comple-tion (±SD) | option (±SD) | comple-tion (±SD) | option (±SD) | comple-tion (±SD) | option (±SD) | comple-tion | option |
| gemma-2b | 0.437 | 0.197 | 0.437 (±0.005) | 0.248 (±0.005) | 0.524 (±0.005) | 0.276 (±0.007) | 0.572 (±0.004) | 0.267 (±0.011) | 0.732 | 0.259 |
| gemma-1.1-2b-it | 0.388 | 0.264 | 0.428 (±0.005) | 0.347 (±0.004) | 0.526 (±0.009) | 0.357 (±0.008) | 0.540 (±0.008) | 0.336 (±0.004) | 0.750 | 0.007 |
| gemma-7b | 0.454 | 0.388 | 0.465 (±0.001) | 0.631 (±0.012) | 0.561 (±0.002) | 0.691 (±0.010) | 0.620 (±0.010) | 0.720 (±0.006) | 0.797 | 0.722 |
| gemma-1.1-7b-it | 0.663 | 0.645 | 0.689 (±0.009) | 0.684 (±0.006) | 0.730 (±0.007) | 0.714 (±0.003) | 0.759 (±0.007) | 0.707 (±0.004) | 0.786 | 0.733 |
| Llama-3.2-3B | 0.444 | 0.313 | 0.481 (±0.006) | 0.417 (±0.011) | 0.574 (±0.006) | 0.463 (±0.009) | 0.601 (±0.010) | 0.501 (±0.009) | 0.733 | 0.342 |
| Llama-3.2-3B-Instruct | 0.472 | 0.530 | 0.487 (±0.009) | 0.540 (±0.011) | 0.557 (±0.006) | 0.526 (±0.006) | 0.587 (±0.004) | 0.532 (±0.005) | 0.745 | 0.604 |
| Llama-3.1-8B | 0.439 | 0.459 | 0.493 (±0.007) | 0.549 (±0.012) | 0.596 (±0.006) | 0.635 (±0.008) | 0.635 (±0.006) | 0.667 (±0.011) | 0.797 | 0.556 |
| Llama-3.1-8B-Instruct | 0.512 | 0.744 | 0.571 (±0.007) | 0.708 (±0.007) | 0.646 (±0.007) | 0.726 (±0.008) | 0.675 (±0.004) | 0.729 (±0.011) | 0.756 | 0.747 |
| Mistral-7B-v0.3 | 0.425 | 0.376 | 0.471 (±0.003) | 0.603 (±0.008) | 0.548 (±0.012) | 0.702 (±0.010) | 0.597 (±0.002) | 0.720 (±0.013) | 0.773 | 0.386 |
| Mistral-7B-Instruct-v0.3 | 0.619 | 0.642 | 0.635 (±0.007) | 0.630 (±0.007) | 0.664 (±0.005) | 0.664 (±0.004) | 0.690 (±0.008) | 0.667 (±0.007) | 0.765 | 0.650 |
| Qwen2.5-3B | 0.458 | 0.500 | 0.470 (±0.008) | 0.538 (±0.008) | 0.520 (±0.005) | 0.560 (±0.008) | 0.554 (±0.006) | 0.567 (±0.009) | 0.724 | 0.629 |
| Qwen2.5-3B-Instruct | 0.534 | 0.620 | 0.599 (±0.007) | 0.584 (±0.003) | 0.633 (±0.009) | 0.600 (±0.007) | 0.656 (±0.007) | 0.609 (±0.006) | 0.700 | 0.726 |
| Qwen2.5-7B | 0.480 | 0.623 | 0.474 (±0.008) | 0.608 (±0.007) | 0.504 (±0.004) | 0.622 (±0.007) | 0.556 (±0.006) | 0.629 (±0.003) | 0.761 | 0.656 |
| Qwen2.5-7B-Instruct | 0.542 | 0.676 | 0.558 (±0.007) | 0.637 (±0.005) | 0.620 (±0.004) | 0.623 (±0.005) | 0.659 (±0.005) | 0.614 (±0.005) | 0.718 | 0.793 |
| gpt-4o-mini | - | 0.643 | - | 0.682 (±0.002) | - | 0.687 (±0.005) | - | 0.704 (±0.008) | - | - |
| gpt-4.1-mini | - | 0.797 | - | 0.822 (±0.005) | - | 0.841 (±0.008) | - | 0.846 (±0.005) | - | - |
| claude-3-5-haiku-latest | - | 0.722 | - | 0.505 (±0.016) | - | 0.476 (±0.009) | - | 0.633 (±0.004) | - | - |
| 2-3B models average | 0.456 | 0.404 | 0.484 | 0.446 | 0.556 | 0.464 | 0.585 | 0.469 | 0.731 | 0.428 |
| 7-8B models average | 0.517 | 0.569 | 0.545 | 0.631 | 0.609 | 0.672 | 0.649 | 0.682 | 0.769 | 0.655 |
| api models average | - | 0.721 | - | 0.670 | - | 0.668 | - | 0.728 | - | - |
| base models average | 0.448 | 0.408 | 0.470 | 0.513 | 0.547 | 0.564 | 0.591 | 0.582 | 0.760 | 0.507 |
| instruct-tuned models average | 0.533 | 0.589 | 0.567 | 0.59 | 0.625 | 0.601 | 0.652 | 0.599 | 0.746 | 0.608 |
| total average | 0.491 | 0.538 | 0.518 | 0.573 | 0.586 | 0.598 | 0.622 | 0.615 | 0.753 | 0.558 |

Table 13: Zero-shot, few-shot, and SFT evaluation results on NUBench with the **definition prompt.** SD denotes standard deviation across random seeds or runs. Few-shot results are averaged over five random seeds (42, 1234, 3000, 5000, and 7000) and one, five, and ten demonstration examples (1-, 5-, 10-shot). Red text indicates the model with the highest performance in each column (excluding API models).

| evaluation setting | zero-shot | | 1-shot | | 5-shot | | 10-shot | | After SFT | |
| --- | --- | --- | --- | --- | --- | --- | --- | --- | --- | --- |
| | comple-tion | option | comple-tion (±SD) | option (±SD) | comple-tion (±SD) | option (±SD) | comple-tion (±SD) | option (±SD) | comple-tion | option |
| gemma-2b | 0.443 | 0.176 | 0.427 (±0.002) | 0.260 (±0.002) | 0.515 (±0.005) | 0.272 (±0.009) | 0.560 (±0.007) | 0.266 (±0.014) | 0.720 | 0.259 |
| gemma-1.1-2b-it | 0.402 | 0.263 | 0.436 (±0.004) | 0.322 (±0.006) | 0.542 (±0.006) | 0.319 (±0.004) | 0.570 (±0.005) | 0.309 (±0.007) | 0.753 | 0.334 |
| gemma-7b | 0.501 | 0.404 | 0.484 (±0.006) | 0.568 (±0.009) | 0.581 (±0.008) | 0.638 (±0.008) | 0.630 (±0.002) | 0.675 (±0.006) | 0.803 | 0.709 |
| gemma-1.1-7b-it | 0.656 | 0.614 | 0.680 (±0.006) | 0.627 (±0.009) | 0.728 (±0.006) | 0.657 (±0.006) | 0.748 (±0.005) | 0.668 (±0.006) | 0.775 | 0.692 |
| Llama-3.2-3B | 0.416 | 0.324 | 0.471 (±0.006) | 0.381 (±0.008) | 0.561 (±0.007) | 0.451 (±0.008) | 0.590 (±0.008) | 0.491 (±0.009) | 0.754 | 0.282 |
| Llama-3.2-3B-Instruct | 0.454 | 0.516 | 0.477 (±0.006) | 0.464 (±0.010) | 0.546 (±0.005) | 0.452 (±0.008) | 0.589 (±0.004) | 0.450 (±0.006) | 0.769 | 0.386 |
| Llama-3.1-8B | 0.462 | 0.477 | 0.499 (±0.005) | 0.512 (±0.013) | 0.598 (±0.006) | 0.591 (±0.008) | 0.645 (±0.005) | 0.630 (±0.009) | 0.817 | 0.608 |
| Llama-3.1-8B-Instruct | 0.519 | 0.684 | 0.557 (±0.005) | 0.640 (±0.004) | 0.628 (±0.002) | 0.663 (±0.004) | 0.665 (±0.007) | 0.681 (±0.014) | 0.784 | 0.762 |
| Mistral-7B-v0.3 | 0.446 | 0.297 | 0.475 (±0.006) | 0.571 (±0.013) | 0.560 (±0.011) | 0.683 (±0.010) | 0.608 (±0.002) | 0.694 (±0.007) | 0.781 | 0.427 |
| Mistral-7B-Instruct-v0.3 | 0.596 | 0.630 | 0.635 (±0.006) | 0.622 (±0.007) | 0.685 (±0.005) | 0.630 (±0.011) | 0.707 (±0.009) | 0.632 (±0.007) | 0.765 | 0.629 |
| Qwen2.5-3B | 0.427 | 0.531 | 0.474 (±0.007) | 0.490 (±0.007) | 0.540 (±0.003) | 0.499 (±0.006) | 0.574 (±0.007) | 0.500 (±0.007) | 0.722 | 0.648 |
| Qwen2.5-3B-Instruct | 0.474 | 0.585 | 0.530 (±0.008) | 0.511 (±0.006) | 0.628 (±0.007) | 0.512 (±0.006) | 0.665 (±0.010) | 0.513 (±0.006) | 0.696 | 0.713 |
| Qwen2.5-7B | 0.454 | 0.628 | 0.483 (±0.009) | 0.606 (±0.005) | 0.541 (±0.007) | 0.622 (±0.007) | 0.587 (±0.002) | 0.639 (±0.007) | 0.745 | 0.706 |
| Qwen2.5-7B-Instruct | 0.536 | 0.636 | 0.549 (±0.007) | 0.634 (±0.009) | 0.634 (±0.010) | 0.634 (±0.008) | 0.674 (±0.006) | 0.636 (±0.004) | 0.734 | 0.764 |
| gpt-4o-mini | - | 0.621 | - | 0.659 (±0.008) | - | 0.688 (±0.005) | - | 0.700 (±0.009) | - | - |
| gpt-4.1-mini | - | 0.842 | - | 0.845 (±0.007) | - | 0.857 (±0.005) | - | 0.867 (±0.005) | - | - |
| claude-3-5-haiku-latest | - | 0.688 | - | 0.495 (±0.006) | - | 0.470 (±0.008) | - | 0.613 (±0.013) | - | - |
| 2-3B models average | 0.436 | 0.399 | 0.469 | 0.405 | 0.555 | 0.418 | 0.591 | 0.422 | 0.736 | 0.437 |
| 7-8B models average | 0.521 | 0.546 | 0.545 | 0.598 | 0.619 | 0.640 | 0.658 | 0.657 | 0.775 | 0.662 |
| api models average | - | 0.717 | - | 0.666 | - | 0.672 | - | 0.727 | - | - |
| base models average | 0.450 | 0.405 | 0.473 | 0.484 | 0.557 | 0.537 | 0.599 | 0.556 | 0.763 | 0.520 |
| instruct-tuned models average | 0.520 | 0.561 | 0.552 | 0.546 | 0.627 | 0.552 | 0.660 | 0.556 | 0.754 | 0.611 |
| total average | 0.485 | 0.524 | 0.513 | 0.542 | 0.592 | 0.567 | 0.629 | 0.586 | 0.758 | 0.566 |

Table 14: Zero-shot, few-shot, and SFT evaluation results on NUBench with the **detail prompt.** SD denotes standard deviation across random seeds or runs. Few-shot results are averaged over five random seeds (42, 1234, 3000, 5000, and 7000) and one, five, and ten demonstration examples (1-, 5-, 10-shot). Red text indicates the model with the highest performance in each column (excluding API models).

| evaluation setting | zero-shot | | 1-shot | | 5-shot | | 10-shot | | After SFT | |
|---|---|---|---|---|---|---|---|---|---|---|
| | comple-tion | option | comple-tion (±SD) | option (±SD) | comple-tion (±SD) | option (±SD) | comple-tion (±SD) | option (±SD) | comple-tion | option |
| gemma-2b | 0.404 | 0.257 | 0.423 (±0.003) | 0.253 (±0.005) | 0.511 (±0.003) | 0.257 (±0.012) | 0.547 (±0.004) | 0.145 (±0.009) | 0.496 | 0.514 |
| gemma-1.1-2b-it | 0.394 | 0.260 | 0.440 (±0.007) | 0.282 (±0.005) | 0.524 (±0.009) | 0.241 (±0.007) | 0.561 (±0.009) | 0.242 (±0.009) | 0.519 | 0.476 |
| gemma-7b | 0.462 | 0.410 | 0.472 (±0.005) | 0.483 (±0.008) | 0.564 (±0.004) | 0.531 (±0.007) | 0.607 (±0.010) | 0.549 (±0.008) | 0.731 | 0.635 |
| gemma-1.1-7b-it | 0.588 | 0.472 | 0.653 (±0.008) | 0.510 (±0.009) | 0.708 (±0.009) | 0.501 (±0.004) | 0.728 (±0.006) | 0.533 (±0.011) | 0.725 | 0.616 |
| Llama-3.2-3B | 0.445 | 0.306 | 0.463 (±0.005) | 0.317 (±0.011) | 0.550 (±0.008) | 0.365 (±0.010) | 0.582 (±0.006) | 0.407 (±0.008) | 0.567 | 0.553 |
| Llama-3.2-3B-Instruct | 0.454 | 0.429 | 0.440 (±0.006) | 0.387 (±0.009) | 0.501 (±0.007) | 0.381 (±0.008) | 0.560 (±0.005) | 0.389 (±0.006) | 0.586 | 0.595 |
| Llama-3.1-8B | 0.442 | 0.402 | 0.466 (±0.004) | 0.394 (±0.008) | 0.566 (±0.006) | 0.462 (±0.012) | 0.624 (±0.005) | 0.518 (±0.010) | 0.726 | 0.642 |
| Llama-3.1-8B-Instruct | 0.508 | 0.539 | 0.529 (±0.008) | 0.515 (±0.010) | 0.627 (±0.005) | 0.554 (±0.004) | 0.669 (±0.003) | 0.580 (±0.006) | 0.673 | 0.681 |
| Mistral-7B-v0.3 | 0.454 | 0.236 | 0.485 (±0.004) | 0.416 (±0.014) | 0.551 (±0.010) | 0.597 (±0.009) | 0.591 (±0.007) | 0.631 (±0.007) | 0.586 | 0.545 |
| Mistral-7B-Instruct-v0.3 | 0.588 | 0.586 | 0.608 (±0.013) | 0.551 (±0.010) | 0.662 (±0.005) | 0.556 (±0.008) | 0.685 (±0.005) | 0.558 (±0.005) | 0.672 | 0.646 |
| Qwen2.5-3B | 0.415 | 0.463 | 0.461 (±0.005) | 0.399 (±0.002) | 0.532 (±0.005) | 0.324 (±0.006) | 0.568 (±0.006) | 0.313 (±0.003) | 0.662 | 0.615 |
| Qwen2.5-3B-Instruct | 0.467 | 0.548 | 0.492 (±0.009) | 0.473 (±0.009) | 0.653 (±0.008) | 0.227 (±0.008) | 0.671 (±0.008) | 0.495 (±0.011) | 0.674 | 0.572 |
| Qwen2.5-7B | 0.443 | 0.555 | 0.464 (±0.005) | 0.515 (±0.010) | 0.537 (±0.008) | 0.542 (±0.005) | 0.577 (±0.003) | 0.566 (±0.004) | 0.691 | 0.689 |
| Qwen2.5-7B-Instruct | 0.507 | 0.545 | 0.511 (±0.006) | 0.527 (±0.008) | 0.621 (±0.012) | 0.526 (±0.007) | 0.662 (±0.004) | 0.544 (±0.003) | 0.674 | 0.705 |
| gpt-4o-mini | - | 0.595 | - | 0.620 (±0.014) | - | 0.647 (±0.003) | - | 0.669 (±0.006) | - | - |
| gpt-4.1-mini | - | 0.858 | - | 0.864 (±0.004) | - | 0.876 (±0.003) | - | 0.870 (±0.005) | - | - |
| claude-3-5-haiku-latest | - | 0.604 | - | 0.472 (±0.009) | - | 0.537 (±0.009) | - | 0.689 (±0.013) | - | - |
| 2-3B models average | 0.430 | 0.377 | 0.453 | 0.352 | 0.545 | 0.299 | 0.582 | 0.332 | 0.584 | 0.554 |
| 7-8B models average | 0.499 | 0.468 | 0.524 | 0.489 | 0.605 | 0.534 | 0.643 | 0.560 | 0.685 | 0.645 |
| api models average | - | 0.686 | - | 0.652 | - | 0.687 | - | 0.743 | - | - |
| base models average | 0.438 | 0.376 | 0.462 | 0.397 | 0.544 | 0.440 | 0.585 | 0.447 | 0.637 | 0.599 |
| instruct-tuned models average | 0.501 | 0.483 | 0.525 | 0.464 | 0.614 | 0.427 | 0.648 | 0.477 | 0.646 | 0.613 |
| total average | 0.469 | 0.474 | 0.493 | 0.469 | 0.579 | 0.478 | 0.617 | 0.512 | 0.642 | 0.606 |

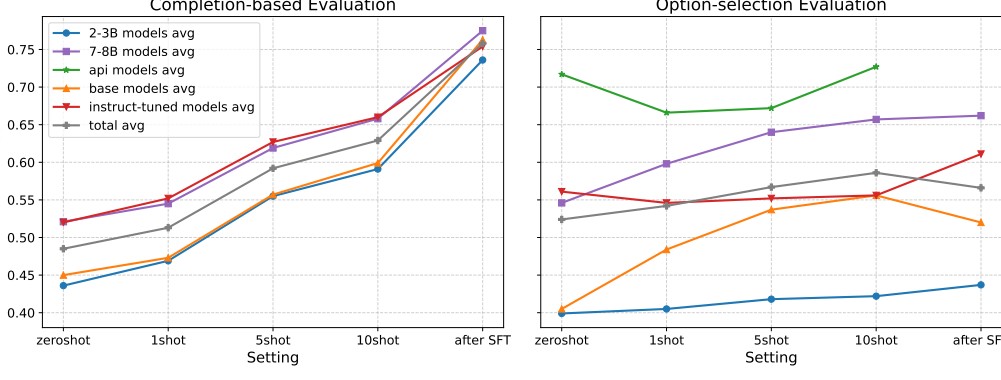

Figure 4: Model performance on NUBench with definition prompt. Circles (blue) represent the average performance of 2-3B models, squares (purple) indicate the average for 7-8B models, upward triangles (orange) signify the average of base models, and downward triangles (red) denote the average of instruction-tuned models. Stars (green) represent API models.

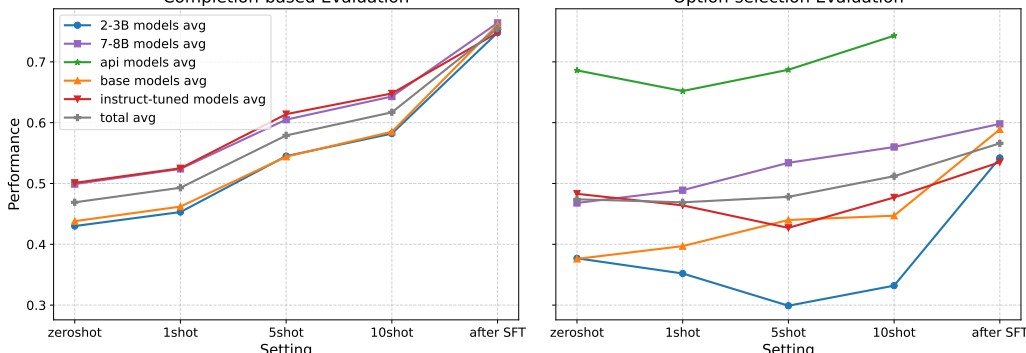

Figure 5: Model performance on NUBench with detail prompt. Circles (blue) represent the average performance of 2-3B models, squares (purple) indicate the average for 7-8B models, upward triangles (orange) signify the average of base models, and downward triangles (red) denote the average of instruction-tuned models. Stars (green) represent API models.

- **Definition prompts.** Performance patterns closely resemble those of simple prompts. However, in the option-selection evaluation, 2-3B models show smaller gains from additional shots, while 7-8B models tend to achieve slightly higher accuracy after SFT compared to 10-shot prompting.

- **Detail prompts.** For smaller (2-3B) models in the option-selection evaluation, adding more shots can actually reduce performance, reflecting difficulties with long and complex prompt instructions to these small models. Nonetheless, these models, along with larger ones, show larger SFT gains under detail prompts than under simple or definition prompts.

Overall, these findings show that prompt formulation plays a non-trivial role in shaping performance trends. In all cases, SFT consistently boosts zero-shot performance.

## Q  GENERAL BENCHMARK PERFORMANCE AFTER SFT

To assess whether supervised fine-tuning (SFT) on NUBench affects performance on broader natural language understanding tasks, we evaluate models on six widely used benchmarks: ARC-Challenge, ARC-Easy, GSM8K, HellaSwag, OpenBookQA, and WinoGrande (Clark et al., 2018; Cobbe et al.,

Table 15: Average performance of 2–3B / 7–8B, and base / instruction-tuned models on six general benchmarks before and after SFT on NUBench. Here, *acc* denotes accuracy, *acc_norm* is normalized accuracy, and *exact_match* requires an exact string match with the reference answer.

| | tasks | arc_challenge | arc_easy | gsm8k | hellaswag | openbook-qa | wino-grande | avg |
|---|---|---|---|---|---|---|---|---|
| | metric | acc | acc | exact_match | acc_norm | acc_norm | acc | |
| **Before SFT with simple prompt on NUBench** | 2-3B models average | 0.432 | 0.747 | 0.325 | 0.714 | 0.409 | 0.668 | **0.549** |
| | 7-8B models average | 0.514 | 0.810 | 0.592 | 0.797 | 0.458 | 0.727 | **0.65** |
| | base models average | 0.464 | 0.784 | 0.480 | 0.768 | 0.437 | 0.709 | **0.607** |
| | instruct-tuned models average | 0.493 | 0.782 | 0.475 | 0.756 | 0.437 | 0.694 | **0.606** |
| | total average | **0.479** | **0.783** | **0.478** | **0.762** | **0.437** | **0.702** | 0.607 |
| **After SFT with simple prompt on NUBench** | 2-3B models average | 0.431 | 0.746 | 0.365 | 0.713 | 0.411 | 0.671 | **0.556** |
| | 7-8B models average | 0.524 | 0.819 | 0.536 | 0.798 | 0.453 | 0.721 | **0.642** |
| | base models average | 0.474 | 0.788 | 0.428 | 0.772 | 0.437 | 0.706 | **0.601** |
| | instruct-tuned models average | 0.495 | 0.787 | 0.497 | 0.751 | 0.432 | 0.693 | **0.609** |
| | total average | **0.484** | **0.788** | **0.463** | **0.762** | **0.435** | **0.699** | 0.605 |
| **Before SFT with definition prompt on NUBench** | 2-3B models average | 0.431 | 0.747 | 0.325 | 0.714 | 0.409 | 0.668 | **0.549** |
| | 7-8B models average | 0.514 | 0.810 | 0.592 | 0.797 | 0.458 | 0.727 | **0.65** |
| | base models average | 0.464 | 0.784 | 0.480 | 0.768 | 0.437 | 0.709 | **0.607** |
| | instruct-tuned models average | 0.493 | 0.782 | 0.475 | 0.755 | 0.437 | 0.694 | **0.606** |
| | total average | **0.479** | **0.783** | **0.478** | **0.762** | **0.437** | **0.701** | 0.607 |
| **After SFT with definition prompt on NUBench** | 2-3B models average | 0.433 | 0.745 | 0.341 | 0.713 | 0.407 | 0.669 | **0.551** |
| | 7-8B models average | 0.528 | 0.819 | 0.545 | 0.799 | 0.455 | 0.725 | **0.645** |
| | base models average | 0.477 | 0.785 | 0.419 | 0.772 | 0.436 | 0.709 | **0.600** |
| | instruct-tuned models average | 0.498 | 0.790 | 0.496 | 0.753 | 0.433 | 0.694 | **0.611** |
| | total average | **0.487** | **0.788** | **0.457** | **0.762** | **0.434** | **0.701** | 0.605 |
| **Before SFT with detail prompt on NUBench** | 2-3B models average | 0.431 | 0.747 | 0.325 | 0.714 | 0.409 | 0.668 | **0.549** |
| | 7-8B models average | 0.514 | 0.810 | 0.592 | 0.797 | 0.458 | 0.727 | **0.65** |
| | base models average | 0.464 | 0.784 | 0.480 | 0.768 | 0.437 | 0.709 | **0.607** |
| | instruct-tuned models average | 0.493 | 0.782 | 0.475 | 0.755 | 0.437 | 0.694 | **0.606** |
| | total average | **0.479** | **0.783** | **0.478** | **0.762** | **0.437** | **0.701** | 0.607 |
| **After SFT with detail prompt on NUBench** | 2-3B models average | 0.429 | 0.746 | 0.359 | 0.713 | 0.409 | 0.668 | **0.554** |
| | 7-8B models average | 0.523 | 0.816 | 0.551 | 0.8 | 0.458 | 0.723 | **0.645** |
| | base models average | 0.469 | 0.785 | 0.424 | 0.773 | 0.439 | 0.704 | **0.599** |
| | instruct-tuned models average | 0.496 | 0.787 | 0.513 | 0.753 | 0.434 | 0.695 | **0.613** |
| | total average | **0.483** | **0.786** | **0.469** | **0.763** | **0.437** | **0.699** | 0.606 |

2021; Zellers et al., 2019; Mihaylov et al., 2018; Sakaguchi et al., 2021). These datasets cover a diverse range of domains, including commonsense reasoning, scientific knowledge, mathematics, and reading comprehension.

Following the classification of prompts introduced in Appendix O, SFT is conducted separately for the simple, definition, and detail prompt styles. We then report post-SFT zero-shot performance across the six benchmarks to verify whether task-specific training on NUBench preserves general capabilities.

Table 15 summarizes the results. We find that performance on general benchmarks remains broadly stable after SFT, indicating that fine-tuning on NUBench does not substantially harm general capabilities.

# R    TOTAL ANALYSIS OF MODEL PREDICTIONS ON THE NUBENCH

This appendix extends the error analysis presented in Section 5.3, providing the complete results. In particular, we examine (i) incorrect choice distributions and (ii) confusion rates for local negation categories, comparing models of different sizes (2-3B vs. 7-8B) and training paradigms (pretrained vs. instruction-tuned) under both zero-shot and few-shot conditions.

We report few-shot results using a fixed random seed (1234), which corresponds to the default seed used in the LM Evaluation Harness framework. Averaging over multiple seeds was avoided, as it could obscure specific error patterns and make fine-grained confusion analysis less interpretable.

For consistency with the main text, we report results only under the simple prompt, which serves as the default evaluation setting throughout the paper.

We organize the complete prediction analysis by model family.

- Results for the Gemma family are reported in Table 16, Table 17.
- Results for the LLaMA family are reported in Table 18, Table 19.
- Results for the Mistral family are reported in Table 20, Table 21.
- Results for the Qwen family are reported in Table 22, Table 23.
- Results for API models are reported in Table 24.

Each table follows the same format, reporting error rates, incorrect choice distributions, and confusion rates across local negation subcategories under zero-shot, few-shot, and SFT conditions.

In the option-selection evaluation setting, we also track cases labeled as "Answer Format Wrong." This category captures instances where the model's output does not follow the required answer format (selecting strictly one of A, B, C, or D). Because such responses cannot be mapped to a specific incorrect option, they are not included in the incorrect choice distribution or confusion rate. Instead, we report their raw counts alongside the other error statistics. This also serves as an indicator of the model's ability to follow output-format instructions.

Overall, we observe consistent trends that larger models (7-8B) achieve lower error rates than smaller ones (2-3B), and introduction-tuned variants generally outperform base models. Supervised fine-tuning (SFT) tends to reduce error rates. Meanwhile, incorrect choice distributions and local negation confusion rates vary across models and settings, showing that this analysis can serve as a useful tool to identify which aspects of negation remain particularly challenging.

Notably, we also find some unusual behaviors. For example, the Gemma-1.1-2B-it model after SFT produces 1,239 outputs with incorrect answer formats in the option-selection setting, indicating formatting issues rather than genuine reasoning errors. Similarly, for API models, Claude 3.5 Haiku shows increasing formatting errors as more shots are added, which directly degrades performance.

Table 16: Error rates, incorrect choice distributions, and local negation confusion rates for the **Gemma** family under zero-shot, few-shot, and SFT conditions, evaluated in the **completion-based setting.**

| | | | Error Rate (1-acc) | Incorrect Choice Distribution | | | Local Negation Confusion Rate | | | |
| --- | --- | --- | --- | --- | --- | --- | --- | --- | --- | --- |
| | | | | Local Negation (%) | Contra-diction (%) | Para-phrase (%) | Relative Clause (%) | Participle Clause (%) | Compound Sentence (%) | Adverbial Clause (%) |
| gemma-2b | baseline | zero-shot | 0.563 | 74.08 | 21.41 | 4.51 | 29.17 | 32.14 | 67.69 | 44.19 |
| | | 1-shot | 0.558 | 74.54 | 21.34 | 4.13 | 32.37 | 31.49 | 63.27 | 45.16 |
| | | 5-shot | 0.474 | 81.61 | 15.89 | 2.51 | 26.28 | 30.19 | 55.78 | 48.06 |
| | | 10-shot | 0.421 | 82.49 | 16.01 | 1.51 | 23.72 | 23.38 | 48.64 | 48.06 |
| | sft | zero-shot | 0.268 | 84.91 | 14.20 | 0.89 | 13.14 | 18.51 | 26.87 | 35.48 |
| gemma-1.1-2b-it | baseline | zero-shot | 0.612 | 59.20 | 32.25 | 8.55 | 31.09 | 28.25 | 55.44 | 35.48 |
| | | 1-shot | 0.565 | 66.76 | 28.47 | 4.77 | 32.37 | 29.55 | 55.10 | 39.35 |
| | | 5-shot | 0.471 | 66.50 | 27.44 | 6.06 | 25.64 | 24.35 | 47.96 | 31.94 |
| | | 10-shot | 0.468 | 68.98 | 26.44 | 4.58 | 26.28 | 22.40 | 49.66 | 35.48 |
| | sft | zero-shot | 0.250 | 81.27 | 17.78 | 0.95 | 11.86 | 15.26 | 26.87 | 30.00 |
| gemma-7b | baseline | zero-shot | 0.546 | 72.28 | 24.24 | 3.48 | 29.49 | 32.14 | 59.18 | 42.90 |
| | | 1-shot | 0.535 | 77.74 | 18.10 | 4.15 | 30.77 | 31.82 | 57.82 | 51.61 |
| | | 5-shot | 0.437 | 84.39 | 13.79 | 1.81 | 25.64 | 26.95 | 47.96 | 51.94 |
| | | 10-shot | 0.369 | 84.95 | 13.33 | 1.72 | 22.44 | 23.05 | 35.71 | 48.06 |
| | sft | zero-shot | 0.203 | 81.64 | 16.80 | 1.56 | 10.26 | 11.36 | 18.03 | 28.71 |
| gemma-1.1-7b-it | baseline | zero-shot | 0.337 | 55.06 | 38.35 | 6.59 | 9.29 | 14.29 | 26.53 | 26.77 |
| | | 1-shot | 0.310 | 69.57 | 26.09 | 4.35 | 14.10 | 13.31 | 26.53 | 35.16 |
| | | 5-shot | 0.279 | 73.01 | 25.00 | 1.99 | 13.14 | 13.64 | 20.07 | 37.10 |
| | | 10-shot | 0.232 | 73.72 | 24.91 | 1.37 | 10.58 | 7.79 | 15.31 | 36.77 |
| | sft | zero-shot | 0.214 | 80.74 | 17.78 | 1.48 | 10.90 | 11.04 | 17.35 | 31.94 |

Table 17: Error rates, incorrect choice distributions, and local negation confusion rates for the **Gemma** family under zero-shot, few-shot, and SFT conditions, evaluated in the **option-selection setting.**

| | | | Error Rate (1-acc) | Answer Format Wrong | Incorrect Choice Distribution | | | Local Negation Confusion Rate | | | |
| --- | --- | --- | --- | --- | --- | --- | --- | --- | --- | --- | --- |
| | | | | | Local Negation (%) | Contra-diction (%) | Para-phrase (%) | Relative Clause (%) | Participle Clause (%) | Compound Sentence (%) | Adverbial Clause (%) |
| gemma-2b | baseline | zero-shot | 0.803 | 274 | 33.29 | 33.42 | 33.29 | 20.51 | 23.05 | 21.43 | 15.48 |
| | | 1-shot | 0.750 | 0 | 31.18 | 29.6 | 39.22 | 22.44 | 26.62 | 23.47 | 23.87 |
| | | 5-shot | 0.722 | 0 | 30.63 | 23.82 | 45.55 | 20.19 | 23.70 | 28.23 | 19.35 |
| | | 10-shot | 0.736 | 0 | 29.42 | 22.74 | 47.84 | 21.47 | 23.70 | 23.81 | 20.32 |
| | sft | zero-shot | 0.741 | 6 | 32.54 | 34.48 | 32.97 | 22.44 | 27.92 | 25.51 | 22.90 |
| gemma-1.1-2b-it | baseline | zero-shot | 0.738 | 4 | 34.23 | 27.86 | 37.90 | 24.04 | 28.25 | 25.85 | 25.48 |
| | | 1-shot | 0.654 | 0 | 46.67 | 18.91 | 34.42 | 29.81 | 35.71 | 30.61 | 29.68 |
| | | 5-shot | 0.631 | 0 | 48.68 | 14.59 | 36.73 | 32.05 | 37.34 | 27.55 | 29.35 |
| | | 10-shot | 0.660 | 0 | 43.87 | 11.06 | 45.07 | 29.49 | 34.42 | 25.85 | 29.35 |
| | sft | zero-shot | 0.994 | 1,239 | 85.71 | 0 | 14.29 | 0.96 | 0.97 | 1.02 | 0.97 |
| gemma-7b | baseline | zero-shot | 0.612 | 195 | 57.89 | 9.88 | 32.24 | 17.31 | 19.81 | 44.90 | 28.06 |
| | | 1-shot | 0.351 | 0 | 69.07 | 12.19 | 18.74 | 20.51 | 14.61 | 40.82 | 24.84 |
| | | 5-shot | 0.307 | 0 | 77.52 | 15.50 | 6.98 | 22.76 | 17.86 | 35.03 | 22.90 |
| | | 10-shot | 0.288 | 0 | 81.54 | 13.50 | 4.96 | 24.36 | 16.23 | 31.29 | 25.16 |
| | sft | zero-shot | 0.278 | 1 | 70.57 | 12.57 | 16.86 | 11.54 | 14.29 | 28.91 | 26.45 |
| gemma-1.1-7b-it | baseline | zero-shot | 0.355 | 0 | 64.51 | 20.09 | 15.40 | 16.99 | 15.26 | 40.14 | 22.90 |
| | | 1-shot | 0.307 | 0 | 63.05 | 15.76 | 21.19 | 16.03 | 11.36 | 34.01 | 19.03 |
| | | 5-shot | 0.283 | 0 | 63.87 | 18.77 | 17.37 | 16.99 | 9.42 | 29.59 | 19.03 |
| | | 10-shot | 0.289 | 0 | 65.93 | 17.31 | 16.76 | 19.23 | 8.77 | 30.61 | 20.32 |
| | sft | zero-shot | 0.269 | 2 | 76.26 | 16.91 | 6.820 | 22.12 | 14.61 | 27.21 | 20.32 |

Table 18: Error rates, incorrect choice distributions, and local negation confusion rates for the **LLaMA** family under zero-shot, few-shot, and SFT conditions, evaluated in the **completion-based setting**.

| | | | Error Rate (1-acc) | Incorrect Choice Distribution | | | Local Negation Confusion Rate | | | |
|---|---|---|---|---|---|---|---|---|---|---|
| | | | | Local Negation (%) | Contra-diction (%) | Para-phrase (%) | Relative Clause (%) | Participle Clause (%) | Compound Sentence (%) | Adverbial Clause (%) |
| Llama-3.2-3B | baseline | zero-shot | 0.556 | 70.04 | 23.11 | 6.85 | 24.68 | 30.19 | 62.59 | 44.19 |
| | | 1-shot | 0.519 | 75.99 | 20.49 | 3.52 | 25.32 | 30.19 | 60.54 | 47.42 |
| | | 5-shot | 0.419 | 78.41 | 18.94 | 2.65 | 20.19 | 25.32 | 46.60 | 43.87 |
| | | 10-shot | 0.401 | 80.20 | 16.83 | 2.97 | 22.44 | 24.03 | 43.20 | 43.23 |
| | sft | zero-shot | 0.267 | 82.79 | 15.43 | 1.78 | 14.42 | 17.53 | 25.85 | 33.55 |
| Llama-3.2-3B-Instruct | baseline | zero-shot | 0.528 | 72.67 | 23.87 | 3.45 | 23.72 | 29.55 | 50.68 | 54.84 |
| | | 1-shot | 0.502 | 75.36 | 23.22 | 1.42 | 28.85 | 27.27 | 48.98 | 51.29 |
| | | 5-shot | 0.441 | 76.08 | 22.84 | 1.08 | 25.64 | 26.62 | 39.80 | 46.45 |
| | | 10-shot | 0.410 | 76.79 | 21.86 | 1.35 | 25.64 | 24.68 | 35.03 | 44.52 |
| | sft | zero-shot | 0.255 | 83.80 | 14.95 | 1.25 | 12.82 | 13.31 | 27.55 | 34.52 |
| Llama-3.1-8B | baseline | zero-shot | 0.562 | 70.62 | 21.33 | 8.05 | 25.64 | 30.84 | 64.29 | 43.87 |
| | | 1-shot | 0.507 | 77.78 | 18.15 | 4.07 | 25.96 | 29.87 | 59.18 | 48.39 |
| | | 5-shot | 0.393 | 83.87 | 14.31 | 1.81 | 20.83 | 22.40 | 47.62 | 45.81 |
| | | 10-shot | 0.358 | 84.51 | 13.72 | 1.77 | 20.19 | 23.05 | 40.82 | 41.29 |
| | sft | zero-shot | 0.203 | 85.55 | 12.50 | 1.95 | 8.97 | 11.69 | 18.71 | 32.26 |
| Llama-3.1-8B-Instruct | baseline | zero-shot | 0.488 | 70.57 | 26.18 | 3.25 | 24.36 | 25.97 | 54.76 | 37.74 |
| | | 1-shot | 0.419 | 73.86 | 24.43 | 1.70 | 22.76 | 25.97 | 44.56 | 34.84 |
| | | 5-shot | 0.347 | 80.55 | 18.76 | 0.69 | 18.91 | 19.16 | 39.80 | 37.74 |
| | | 10-shot | 0.329 | 81.20 | 18.31 | 0.48 | 19.55 | 20.45 | 34.01 | 36.45 |
| | sft | zero-shot | 0.244 | 86.04 | 12.34 | 1.62 | 12.18 | 12.66 | 28.23 | 33.87 |

Table 19: Error rates, incorrect choice distributions, and local negation confusion rates for the **LLaMA** family under zero-shot, few-shot, and SFT conditions, evaluated in the **option-selection setting.**

| | | | Error Rate (1-acc) | Answer Format Wrong | Incorrect Choice Distribution | | | Local Negation Confusion Rate | | | |
|---|---|---|---|---|---|---|---|---|---|---|---|
| | | | | | Local Negation (%) | Contra-diction (%) | Para-phrase (%) | Relative Clause (%) | Participle Clause (%) | Compound Sentence (%) | Adverbial Clause (%) |
| Llama-3.2-3B | baseline | zero-shot | 0.688 | 0 | 42.21 | 25.95 | 31.83 | 22.12 | 33.12 | 32.65 | 31.94 |
| | | 1-shot | 0.570 | 0 | 62.59 | 14.74 | 22.67 | 31.09 | 32.14 | 45.58 | 38.71 |
| | | 5-shot | 0.548 | 0 | 74.10 | 14.33 | 11.58 | 34.62 | 35.06 | 50.34 | 47.74 |
| | | 10-shot | 0.500 | 0 | 78.13 | 13.00 | 8.87 | 35.90 | 31.82 | 43.88 | 49.68 |
| | sft | zero-shot | 0.658 | 0 | 39.40 | 33.01 | 27.59 | 23.40 | 31.17 | 27.89 | 24.52 |
| Llama-3.2-3B-Instruct | baseline | zero-shot | 0.474 | 8 | 63.90 | 10.34 | 25.76 | 19.55 | 28.25 | 60.20 | 16.77 |
| | | 1-shot | 0.461 | 0 | 71.60 | 12.56 | 15.83 | 23.40 | 25.97 | 61.22 | 26.77 |
| | | 5-shot | 0.472 | 0 | 74.62 | 10.92 | 14.45 | 29.81 | 28.25 | 63.27 | 25.16 |
| | | 10-shot | 0.467 | 0 | 73.85 | 10.36 | 15.79 | 33.01 | 25.00 | 62.24 | 23.23 |
| | sft | zero-shot | 0.437 | 87 | 61.85 | 7.11 | 31.03 | 11.22 | 18.51 | 43.54 | 21.61 |
| Llama-3.1-8B | baseline | zero-shot | 0.541 | 0 | 47.95 | 6.74 | 45.31 | 23.08 | 24.03 | 41.50 | 19.03 |
| | | 1-shot | 0.454 | 0 | 67.66 | 4.55 | 27.80 | 27.56 | 28.25 | 47.62 | 23.87 |
| | | 5-shot | 0.362 | 0 | 78.95 | 6.58 | 14.47 | 30.45 | 21.75 | 40.82 | 25.16 |
| | | 10-shot | 0.316 | 0 | 82.46 | 6.02 | 11.53 | 30.77 | 21.75 | 31.97 | 23.23 |
| | sft | zero-shot | 0.444 | 48 | 58.79 | 27.34 | 13.87 | 20.83 | 20.13 | 37.41 | 20.65 |
| Llama-3.1-8B-Instruct | baseline | zero-shot | 0.256 | 12 | 76.53 | 15.11 | 8.36 | 16.67 | 18.83 | 19.05 | 23.23 |
| | | 1-shot | 0.290 | 0 | 71.51 | 21.37 | 7.12 | 19.23 | 18.18 | 29.25 | 19.03 |
| | | 5-shot | 0.269 | 0 | 74.04 | 21.83 | 4.13 | 20.83 | 16.88 | 25.17 | 19.35 |
| | | 10-shot | 0.290 | 0 | 80.00 | 14.52 | 5.48 | 25.00 | 20.78 | 26.87 | 22.90 |
| | sft | zero-shot | 0.254 | 31 | 72.32 | 19.03 | 8.65 | 15.06 | 17.86 | 21.09 | 14.52 |

Table 20: Error rates, incorrect choice distributions, and local negation confusion rates for the **Mistral** family under zero-shot, few-shot, and SFT conditions, evaluated in the **completion-based setting**.

| | | | Error Rate | Incorrect Choice Distribution | | | Local Negation Confusion Rate | | | |
| | | | (1-acc) | Local Negation (%) | Contra-diction (%) | Para-phrase (%) | Relative Clause (%) | Participle Clause (%) | Compound Sentence (%) | Adverbial Clause (%) |
|---|---|---|---|---|---|---|---|---|---|---|
| **Mistral-7B-v0.3** | baseline | **zero-shot** | 0.575 | 76.97 | 18.48 | 4.55 | 26.60 | 32.14 | 66.33 | 58.39 |
| | | **1-shot** | 0.532 | 78.39 | 17.59 | 4.02 | 26.92 | 33.12 | 59.52 | 53.23 |
| | | **5-shot** | 0.439 | 83.57 | 14.98 | 1.44 | 22.12 | 25.65 | 51.70 | 52.58 |
| | | **10-shot** | 0.401 | 84.39 | 13.83 | 1.78 | 21.15 | 24.68 | 47.28 | 47.10 |
| | sft | **zero-shot** | 0.227 | 88.81 | 9.09 | 2.10 | 8.65 | 13.31 | 29.93 | 31.61 |
| **Mistral-7B-Instruct-v0.3** | baseline | **zero-shot** | 0.381 | 67.29 | 30.83 | 1.88 | 13.46 | 18.51 | 30.95 | 42.90 |
| | | **1-shot** | 0.365 | 72.61 | 26.09 | 1.30 | 15.38 | 20.45 | 34.69 | 39.03 |
| | | **5-shot** | 0.331 | 72.66 | 26.86 | 0.48 | 14.10 | 15.58 | 31.97 | 37.74 |
| | | **10-shot** | 0.304 | 73.63 | 25.85 | 0.52 | 12.18 | 14.29 | 30.61 | 35.48 |
| | sft | **zero-shot** | 0.236 | 85.52 | 13.80 | 0.67 | 10.26 | 12.01 | 27.21 | 33.87 |

Table 21: Error rates, incorrect choice distributions, and local negation confusion rates for the **Mistral** family under zero-shot, few-shot, and SFT conditions, evaluated in the **option-selection setting**.

| | | | Error Rate | Answer Format Wrong | Incorrect Choice Distribution | | | Local Negation Confusion Rate | | | |
| | | | (1-acc) | | Local Negation (%) | Contra-diction (%) | Para-phrase (%) | Relative Clause (%) | Participle Clause (%) | Compound Sentence (%) | Adverbial Clause (%) |
|---|---|---|---|---|---|---|---|---|---|---|---|
| **Mistral-7B-v0.3** | baseline | **zero-shot** | 0.624 | 351 | 53.21 | 21.56 | 25.23 | 16.99 | 18.18 | 26.53 | 14.52 |
| | | **1-shot** | 0.410 | 0 | 67.12 | 16.63 | 16.25 | 30.77 | 24.68 | 28.91 | 29.03 |
| | | **5-shot** | 0.306 | 0 | 83.16 | 11.92 | 4.92 | 26.28 | 24.35 | 24.49 | 29.68 |
| | | **10-shot** | 0.272 | 0 | 85.42 | 8.45 | 6.12 | 23.40 | 22.73 | 21.09 | 28.39 |
| | sft | **zero-shot** | 0.614 | 0 | 40.05 | 29.46 | 30.49 | 22.44 | 27.92 | 28.57 | 22.58 |
| **Mistral-7B-Instruct-v0.3** | baseline | **zero-shot** | 0.358 | 1 | 65.56 | 25.78 | 8.67 | 27.24 | 23.05 | 27.55 | 18.71 |
| | | **1-shot** | 0.370 | 0 | 61.03 | 34.90 | 4.07 | 27.24 | 20.78 | 26.19 | 19.03 |
| | | **5-shot** | 0.335 | 0 | 68.96 | 29.86 | 1.18 | 28.85 | 19.81 | 26.19 | 20.32 |
| | | **10-shot** | 0.331 | 0 | 71.22 | 27.10 | 1.68 | 28.53 | 22.08 | 25.51 | 20.97 |
| | sft | **zero-shot** | 0.351 | 1 | 67.57 | 21.32 | 11.11 | 23.08 | 23.05 | 33.67 | 18.06 |

Table 22: Error rates, incorrect choice distributions, and local negation confusion rates for the **Qwen** family under zero-shot, few-shot, and SFT conditions, evaluated in the **completion-based setting.**

| | | | Error Rate | Incorrect Choice Distribution | | | Local Negation Confusion Rate | | | |
| | | | (1-acc) | Local Negation (%) | Contra-diction (%) | Para-phrase (%) | Relative Clause (%) | Participle Clause (%) | Compound Sentence (%) | Adverbial Clause (%) |
|---|---|---|---|---|---|---|---|---|---|---|
| Qwen2.5-3B | baseline | zero-shot | 0.542 | 67.69 | 26.02 | 6.29 | 24.68 | 30.19 | 55.10 | 42.26 |
| | | 1-shot | 0.522 | 67.63 | 25.53 | 6.84 | 24.36 | 28.25 | 55.78 | 38.06 |
| | | 5-shot | 0.481 | 70.84 | 23.89 | 5.27 | 25.00 | 28.57 | 50.00 | 37.74 |
| | | 10-shot | 0.437 | 73.14 | 22.14 | 4.72 | 23.40 | 25.32 | 45.58 | 38.06 |
| | sft | zero-shot | 0.276 | 85.34 | 13.22 | 1.44 | 14.74 | 20.78 | 30.95 | 30.97 |
| Qwen2.5-3B-Instruct | baseline | zero-shot | 0.466 | 68.03 | 28.57 | 3.40 | 19.23 | 28.25 | 42.86 | 40.97 |
| | | 1-shot | 0.393 | 67.68 | 28.28 | 4.04 | 16.67 | 20.45 | 42.52 | 30.65 |
| | | 5-shot | 0.355 | 77.01 | 21.88 | 1.12 | 19.55 | 22.40 | 39.46 | 31.94 |
| | | 10-shot | 0.351 | 77.83 | 21.04 | 1.13 | 21.15 | 23.05 | 35.71 | 32.90 |
| | sft | zero-shot | 0.300 | 85.19 | 13.49 | 1.32 | 15.71 | 22.40 | 33.33 | 34.19 |
| Qwen2.5-7B | baseline | zero-shot | 0.520 | 70.73 | 24.54 | 4.73 | 24.68 | 31.17 | 53.06 | 43.55 |
| | | 1-shot | 0.534 | 71.47 | 25.56 | 2.97 | 29.17 | 31.17 | 56.12 | 41.61 |
| | | 5-shot | 0.492 | 71.13 | 25.81 | 3.06 | 24.36 | 28.25 | 51.70 | 40.65 |
| | | 10-shot | 0.447 | 74.07 | 23.98 | 1.95 | 24.04 | 24.68 | 45.58 | 42.58 |
| | sft | zero-shot | 0.240 | 87.75 | 11.59 | 0.66 | 15.06 | 15.91 | 22.45 | 33.23 |
| Qwen2.5-7B-Instruct | baseline | zero-shot | 0.458 | 65.16 | 31.54 | 3.29 | 18.27 | 29.22 | 39.80 | 36.13 |
| | | 1-shot | 0.438 | 69.02 | 30.07 | 0.91 | 22.12 | 23.70 | 43.88 | 35.48 |
| | | 5-shot | 0.378 | 71.85 | 27.94 | 0.21 | 19.55 | 19.16 | 37.41 | 36.13 |
| | | 10-shot | 0.335 | 73.76 | 25.77 | 0.47 | 19.23 | 18.18 | 30.61 | 34.19 |
| | sft | zero-shot | 0.282 | 88.76 | 10.96 | 0.28 | 16.03 | 19.48 | 31.29 | 36.77 |

Table 23: Error rates, incorrect choice distributions, and local negation confusion rates for the **Qwen** family under zero-shot, few-shot, and SFT conditions, evaluated in the **option-selection setting.**

| | | | Error Rate | Answer Format Wrong | Incorrect Choice Distribution | | | Local Negation Confusion Rate | | | |
| | | | (1-acc) | | Local Negation (%) | Contra-diction (%) | Para-phrase (%) | Relative Clause (%) | Participle Clause (%) | Compound Sentence (%) | Adverbial Clause (%) |
|---|---|---|---|---|---|---|---|---|---|---|---|
| Qwen2.5-3B | baseline | zero-shot | 0.500 | 0 | 38.99 | 19.49 | 41.52 | 23.08 | 19.16 | 26.87 | 11.61 |
| | | 1-shot | 0.463 | 0 | 45.89 | 27.23 | 26.88 | 25.00 | 21.43 | 26.53 | 14.84 |
| | | 5-shot | 0.431 | 0 | 44.75 | 25.60 | 29.65 | 24.04 | 19.81 | 23.13 | 12.58 |
| | | 10-shot | 0.439 | 0 | 45.85 | 23.10 | 31.05 | 25.00 | 21.10 | 25.17 | 11.94 |
| | sft | zero-shot | 0.373 | 2 | 51.07 | 13.25 | 35.68 | 17.31 | 15.58 | 26.19 | 19.35 |
| Qwen2.5-3B-Instruct | baseline | zero-shot | 0.380 | 0 | 49.27 | 40.29 | 10.44 | 20.19 | 15.91 | 24.15 | 17.10 |
| | | 1-shot | 0.419 | 0 | 53.22 | 32.39 | 14.39 | 26.28 | 19.16 | 28.23 | 18.39 |
| | | 5-shot | 0.398 | 2 | 54.40 | 31.60 | 14.00 | 27.24 | 19.16 | 29.25 | 13.55 |
| | | 10-shot | 0.388 | 0 | 56.44 | 29.24 | 14.31 | 26.60 | 20.13 | 30.61 | 13.23 |
| | sft | zero-shot | 0.274 | 1 | 59.13 | 34.49 | 6.38 | 12.18 | 13.31 | 21.43 | 20.00 |
| Qwen2.5-7B | baseline | zero-shot | 0.377 | 0 | 68.21 | 15.79 | 16.00 | 28.53 | 22.73 | 36.39 | 18.71 |
| | | 1-shot | 0.392 | 0 | 70.65 | 15.59 | 13.77 | 33.97 | 26.30 | 38.10 | 16.13 |
| | | 5-shot | 0.377 | 0 | 72.00 | 15.16 | 12.84 | 33.65 | 23.05 | 38.78 | 16.77 |
| | | 10-shot | 0.372 | 0 | 73.13 | 14.93 | 11.94 | 32.69 | 23.05 | 40.48 | 16.45 |
| | sft | zero-shot | 0.344 | 0 | 82.03 | 13.13 | 4.84 | 30.77 | 24.35 | 43.54 | 18.39 |
| Qwen2.5-7B-Instruct | baseline | zero-shot | 0.324 | 0 | 62.84 | 29.34 | 7.82 | 23.72 | 22.73 | 23.47 | 14.19 |
| | | 1-shot | 0.363 | 0 | 61.79 | 31.66 | 6.55 | 30.45 | 21.10 | 27.21 | 13.87 |
| | | 5-shot | 0.368 | 0 | 66.38 | 26.51 | 7.11 | 31.41 | 19.81 | 34.69 | 15.16 |
| | | 10-shot | 0.390 | 0 | 67.07 | 25.81 | 7.11 | 34.62 | 20.78 | 35.03 | 17.74 |
| | sft | zero-shot | 0.207 | 0 | 78.54 | 17.62 | 3.83 | 18.59 | 20.13 | 17.69 | 10.65 |

Table 24: Error rates, incorrect choice distributions, and local negation confusion rates for the **API models** under zero-shot and few-shot conditions, evaluated in the **option-selection setting.**

| | | | Error Rate (1-acc) | Answer Format Wrong | Incorrect Choice Distribution | | | Local Negation Confusion Rate | | | |
| --- | --- | --- | --- | --- | --- | --- | --- | --- | --- | --- | --- |
| | | | | | Local Negation (%) | Contra-diction (%) | Para-phrase (%) | Relative Clause (%) | Participle Clause (%) | Compound Sentence (%) | Adverbial Clause (%) |
| gpt-4o-mini | baseline | zero-shot | 0.357 | 0 | 34.67 | 65.33 | 0 | 17.31 | 8.44 | 13.27 | 11.94 |
| | | 1-shot | 0.317 | 0 | 40.00 | 59.75 | 0.25 | 19.87 | 7.47 | 13.95 | 10.97 |
| | | 5-shot | 0.316 | 0 | 38.35 | 61.15 | 0.50 | 19.23 | 7.14 | 11.90 | 11.61 |
| | | 10-shot | 0.298 | 0 | 42.02 | 57.98 | 0 | 19.23 | 8.12 | 12.93 | 11.29 |
| gpt-4.1-mini | baseline | zero-shot | 0.203 | 0 | 51.95 | 48.05 | 0 | 14.10 | 6.49 | 8.16 | 14.52 |
| | | 1-shot | 0.184 | 0 | 64.66 | 35.34 | 0 | 16.35 | 6.17 | 10.20 | 16.13 |
| | | 5-shot | 0.156 | 0 | 65.48 | 34.52 | 0 | 13.46 | 6.17 | 7.14 | 15.16 |
| | | 10-shot | 0.153 | 0 | 64.77 | 35.23 | 0 | 14.74 | 5.19 | 4.76 | 15.81 |
| claude-3-5-haiku-latest | baseline | zero-shot | 0.278 | 0 | 67.81 | 30.48 | 1.71 | 22.76 | 14.29 | 15.99 | 24.52 |
| | | 1-shot | 0.477 | 434 | 83.93 | 15.48 | 0.60 | 20.19 | 6.49 | 10.54 | 8.71 |
| | | 5-shot | 0.527 | 574 | 95.56 | 4.44 | 0 | 13.78 | 3.57 | 5.44 | 5.16 |
| | | 10-shot | 0.368 | 324 | 94.29 | 5.00 | 0.71 | 20.83 | 6.17 | 5.44 | 10.32 |

