# OpenReview forum: "NUBench: A Benchmark for LLMs' Sentence-Level Negation Understanding"
_ICLR.cc/2026/Conference — ICLR 2026 Conference Withdrawn Submission_

### Official Review · Reviewer_mWMq · 2025-10-22

**Soundness:** 2
**Presentation:** 2
**Contribution:** 2
**Rating:** 2
**Confidence:** 4

**Summary:**

This paper introduces NUBench, a benchmark designed to evaluate LLMs understanding of sentence-level negation. The authors provide a formal definition of "standard negation" grounded in sentential logic, create a manually curated dataset with sentence-negation pairs (for finetuning) and multiple-choice questions, and evaluate various LLMs (2-3B to 8B and some proprietary) using both prompting and supervised fine-tuning approaches, and provides detailed error analysis, revealing that models frequently confuse standard negation with local negation.

**Strengths:**

- The dataset is constructed with strong linguistics grounding.

**Weaknesses:**

- The authors provide lots of details on the linguistics grounding and construction process, but failed to state what is the task (and the goal of the benchmark). Figure 1 shows an example. The instruction is simply just "negate the sentence", and the answer is A so does the benchmark only test for standard negation? Also, what if we only negate only one main verbs, e.g. "Batts aren't typically used in the walls"? In my opinion, this sentence is still negated. Please clarify more on what exactly is the benchmark intended to test.

- The claim that NUBench deal with "sentential logic, moving beyond the cue-based and often ambiguous accounts of prior work" is strange. Negation itself modifies sentence-level semantics, and the previous benchmarks are in fact sentence-level. The paper mentions NegNLI, MoNLI, and NaN-NLI but doesn't provide detailed comparisons. If the focus is only on the task framing (generative models), we can easily do that for previous benchmarks too.

- May not have impact beyond the CL/NLP community. The finding that models still struggle with different types of negation is not new.
While the authors included an error analysis, there's insufficient analysis of why models make specific errors.

- Closed model selected are also small variants (mini) instead of the larger, more capable versions.

**Questions:**

- Models perform SFT for the option setting and some models (Gemma-1.1-2b-it) produce 1,239 format errors after SFT. This suggests that SFT is not properly done. Did the authors ensure that the finetuning setting is sound and working as intended? Also, why not SFT using the multiple-choice format?

- Why does option-selection evaluation yield such different patterns from completion-based?

- The fact that simple prompt works better than definition and detailed prompt is questionable. Is it because the samples are very long and contain many main verbs?

---

> ### Author Response · Authors · 2025-11-26
>
> We thank the reviewer for their constructive feedback and for recognizing the strong linguistic grounding of NUBench. We appreciate your insightful questions, which have helped us clarify the paper's core contributions. Below, we address your concerns point-by-point.
>
> **Q1. Unclear task & goal**
> A1. We clarify that the specific goal of NUBench is to evaluate **standard negation**, defined as the logical reversal of the **entire** proposition, rather than merely identifying any linguistically valid form of negation. Regarding your example, we agree that this constitutes a valid negation in a general linguistic sense; however, in our framework, it represents local negation because it negates only a specific constituent while leaving the coordinated clauses true. The benchmark is explicitly designed to test whether models can distinguish this *partial* negation from the *global* negation required to reverse the sentence's overall truth value.
> We acknowledge that the simple prompt "Negate the sentence" allows for this ambiguity. To address this in the revised version, we will revise the main text to primarily highlight results obtained using the Detail Prompt (Appendix O ), which explicitly instructs the model to reverse the truth value of the main predicate. By centering our analysis on this detailed prompt, we will clarify that the benchmark assesses the model's ability to adhere to precise logical scope constraints, rather than its interpretation of a broad instruction.
>
> **Q2. Novelty & Comparison to Prior Work**
> A2. We acknowledge that previous benchmarks also operate at the sentence level; however, the fundamental difference lies in the evaluation objective.
> Benchmarks like NegNLI focus on determining the truth relationship (entailment/contradiction) between two sentences. In these tasks, negation is merely a cue to shift the label. If a model guesses contradiction simply because it sees a negative word, it can often get the right answer without understanding what exactly was negated.
> In contrast, NUBench evaluates the precision of the negation operation itself. We ask which option represents the standard negation (logical reversal) of the sentence. By explicitly including local negation (partial scope) as a distractor, which contains negative words but fails to reverse the entire sentence, NUBench penalizes the shallow heuristics that often succeed in NLI.
> We will expand the "Related Work" section to explicitly contrast these differing objectives (inference versus scope identification) to better highlight NUBench's unique contribution.
>
> **Q3. Limited impact on CL/NLP community. The analysis of why models make specific errors is insufficient.**
> A3. While previous studies established that models fail on dealing with negations, they rarely investigated how these failures distribute across specific syntactic types(standard or local negation) or interact with logical operators. NUBench is unique in quantitatively demonstrating that models do not fail uniformly; they are specifically vulnerable when negation scope interacts with compound structures (e.g., conjunctions) or subordinate clauses.
> We agree that the analysis of why these errors occur can be strengthened. Our current confusion rate analysis (Section 5.3) offers some clues on errors. The high confusion with local negation (especially in compound sentences) suggests that models are performing keyword matching rather than parsing the full logical scope. In the revised version, we will expand the error analysis section.
>
> **Q4. Selection of Closed Models**
> A4. In the revised version, we will include evaluation results for larger models.

---

> ### Author Response · Authors · 2025-11-26
>
> **Q5. SFT Soundness & Training Format**
> A5. Nearly all other models (Llama-3, Mistral, Qwen families) showed performance improvements and adhered to the output format after SFT. This indicates that the training pipeline and hyperparameters were correctly configured.
> The high error count (1,239) in gemma-1.1-2b-it is an isolated anomaly specific to this small-scale model. Our inspection reveals that instead of outputting a single character, the model generated nonsensical tokens after the answer (e.g., "B pimiento", "A vns", "B patrie"). Since our training data (Sentence-Negation pairs) contains no such artifacts, this suggests that the small model tends to forget the instruction-following constraints  (specifically length/format constraints), rather than a flaw in the SFT process itself.
> Considering SFT format, our goal was to teach the model the underlying logic of negation, not merely how to solve a specific multiple-choice exam. Training on the generative task forces the model to internalize the syntactic rules of standard negation.
> Fine-tuning on multiple-choice format mappings(ex. question-option A) risks the model learning superficial heuristics or overfitting to the benchmark's structure. By training on the fundamental task of generating the negated sentence, we aim to improve the model's robust understanding of negation, which is the primary objective of NUBench.
>
> **Q6. Completion-based vs. Option-selection**
> A6. Completion-based evaluation assesses the log-probability (likelihood) of the target sentence directly. In contrast, option-selection is a generative task that requires the model to not only identify the correct answer but also map it to an abstract label (A, B, C, D). This divergence is well-documented in recent literature. For instance, the OLMES standard[1] demonstrates that cloze formulation (completion) and multiple-choice formulation (option-selection) often yield different signals, highlighting the distinction between latent knowledge and generative capability.
> [1] Gu, Yuling, et al. "Olmes: A standard for language model evaluations." Findings of the Association for Computational Linguistics: NAACL 2025. 2025.
>
> **Q7. Effectiveness of Simple Prompts**
> A7. We agree that the syntactic complexity of the samples contributes to the difficulty. However, previous research has pointed out that longer prompts do not strictly guarantee better reasoning and can effectively act as contextual noise rather than helpful guidance[2], which could be the reason of this phenomenon.
> [2] Ye, Xi, and Greg Durrett. "The unreliability of explanations in few-shot prompting for textual reasoning." Advances in neural information processing systems 35 (2022): 30378-30392.

---

### Official Review · Reviewer_g82s · 2025-10-30

**Soundness:** 3
**Presentation:** 3
**Contribution:** 2
**Rating:** 4
**Confidence:** 3

**Summary:**

The paper created a new benchmark for evaluating sentence-level negation understanding in LLMs called "NUBench". The task is a simple multiple-choice question where given a sentence (without negation), the model has to pick the standard negation among the options (local, contradiction, and paraphrase). The paper formalizes standard negation with logical operations and provides a taxonomy on negation (relative clause, participle clause, etc). The results show that the performance on the benchmark increases with more few-shot examples and SFT.

**Strengths:**

* The discussion on operationalizing standard negation across conjunction/disjunction/conditionals is valuable towards understanding negation and consistency.
* Comprehensive evaluation experiments (zero-shot, few-shot, SFT) and inclusion of multiple models from different families, and an error analysis uncovering that more examples result in lower error rates and an unusually high error on selecting the paraphrased option (which doesn't even have negation)
* The creation process is robust with most of the creation being done by authors themselves (standard/local negation), which are challenging for LLMs, and the generated ones are refined by humans (authors).

**Weaknesses:**

1. The inter-annotator agreement and human performance on the benchmark are not reported. Adding the performance of an independent human on the test would help validate the benchmark creation process and better interpret model performances.
2. The related work is broad and doesn't explain how NUBench is different than previous negation benchmarks like CondaQA, LAMA-Neg, NLI-neg, etc.
3. It is not exactly clear to me what this benchmark evaluates and whether there exists a single answer to the question of "negate this sentence". For example, "John went to the shop and bought an apple" can be negated into "John didn't go to the shop or didn't buy an apple" or "John went to the shop but didn't buy an apple". The latter sounds more natural to me, but my understanding is that the benchmarks is looking for the former one. Addressing W.1 should help understand this too. From example in Figure 1, I don't think there's a single way to negate the given sentence in a natural way without more context.
4. LLMs are known to avoid generating/picking unfactual statements because of their safety filters. Given that one of the sources is WikiPedia, there's a high chance that the negated statements are not factual anymore and this would explain the high error rate on picking paraphrases (which should be still factual). Have you verified the factuality of the negated sentences in the dataset? Adding a discussion on this would help understanding errors.

**Questions:**

1. Have you tried prompting LLMs to negate the sentences (without having access to any options) for a subset of NUBench and evaluating the performance and error types?
2. In L.352 you mention that an OpenAI API was used to generate contradiction and paraphrases. Which model was used to do this?

---

> ### Author Response · Authors · 2025-11-26
>
> We thank the reviewer for the constructive feedback and for recognizing the value of our benchmark. We address your specific concerns below:
>
> **Q1. Missing Inter-annotator agreement & Human performance**
> A1. We wish to clarify that NUBench construction involved generative tasks (rewriting/refining sentences) rather than categorical labeling. As such, traditional IAA metrics like Kappa are methodologically inapplicable. Instead, we employed a strict Iterative Consensus Protocol (Appendix I), where every instance was cross-checked and disagreements were adjudicated until unanimous consensus was reached. In the revised version, we will explicitly detail the specific generative nature of each construction step to prevent ambiguity.
> We fully agree that a human baseline can be crucial for validating the benchmark's solvability. Because this task requires human participants, we will conduct a formal human evaluation after obtaining IRB approval.
>
> **Q2. Distinction from Related Work**
> A2. We acknowledge that the unique contributions of NUBench relative to prior work could be made more explicit. In our revised Related Work section, we will clarify that while existing NLI benchmarks (e.g., NegNLI) primarily focus on determining inference relationships (entailment/contradiction) and QA benchmarks (e.g., CondaQA) assess general comprehension of negated contexts, NUBench focuses specifically on identifying the scope of negation. Unlike previous benchmarks, NUBench tests the model's specific ability to distinguish standard negation (logical reversal) from structurally similar local negation (partial scope) and semantic contradiction, demanding a more precise linguistic understanding than broad inference or reading comprehension tasks.
>
> **Q3. Ambiguous Task and Prompts**
> A3. This ambiguity is the core challenge NUBench is designed to evaluate. Our goal is to test if models can adhere to strict logical scope (standard negation) even when local negation seems linguistically plausible.
> We share your concern that the simple prompt "Negate this sentence" might be ambiguous. However, our analysis in Appendix O compared three prompt types: (1) Simple, (2) Definition-based, and (3) Detailed step-by-step instructions. The results showed that explicitly defining standard negation in the prompt did not significantly improve performance compared to the simple prompt.
> To ensure the task definition is unambiguous in the final paper, we will revise the main text to primarily highlight the results obtained using the Detail Prompt. By centering the analysis on the prompt that explicitly outlines the logical scope, we will clarify that the benchmark assesses the model's ability to adhere to precise logical constraints rather than its interpretation of a vague instruction.
>
> **Q4. Factuality and Safety Bias**
> A4. We agree that safety filters and factuality bias might play a role, as models nowadays are more and more optimized to generate factual text. [1]
> However, our completion-based evaluation (Table 7) shows that models frequently generate local negation as the primary error, rather than paraphrases. This suggests the core issue is often scope confusion, not just factuality.
> While paraphrase errors are high in zero-shot option-selection, they decrease significantly after few-shot or SFT. This indicates that SFT successfully re-calibrates the models to prioritize the logical instruction ("Negate") over their pre-trained factuality/safety priors. To further disentangle this in the revised version, we will experiment with adding explicit instructions to the prompt (e.g., "The result need not be factually true") to mitigate any remaining bias.
> [1] Wang, Yuxia, et al. "Factuality of large language models: A survey." Proceedings of the 2024 Conference on Empirical Methods in Natural Language Processing. 2024.

---

> ### Author Response · Authors · 2025-11-26
>
> **Q5. Generative Evaluation**
> A5. We considered generative evaluation during the design phase but concluded that the multiple-choice format is more suitable.
> The additional generative experiment on a subset of NUBench using Qwen3-0.6B with the simple prompt ("Negate the sentence") shows the instability of open-ended generation. Instead of directly generating the negated sentence, the model frequently outputted internal reasoning chains or meta-commentary before (or instead of) the final answer. This inconsistent output format confirms that open-ended generation introduces noise and evaluation overhead compared to the precise, controlled setting of our multiple-choice benchmark. This makes difficult to analyze both performance or the error types.
> Below is the example of the response.
> > Okay, so I need to negate the given sentence. Let me start by understanding what negation means here. Negating a sentence usually involves flipping the meaning completely. So, if the original sentence is saying something true, the negated version would be that it's not true. Looking at the sentence: ""Francis Algernon Disney-Roebuck (1818 or 1819 – 22 March 1885), commonly referred to as Captain Disney-Roebuck or Captain Roebuck, was a British Army officer in Cape Town, South Africa, who became manager of a theatre company of historic importance in South Africa."" First, I'll break down each part. The subject is Francis Algernon Disney-Roebuck. The negation would mean he isn't this person. So maybe ""was not Francis Algernon Disney-Roebuck"" or ""did not exist as Francis Algernon Disney-Roebuck"".
>
> **Q6. Clarification on OpenAI API Models**
> A6. As detailed in Appendix K.2, we used GPT-4o for generating contradictions and paraphrases.

---

### Official Review · Reviewer_3Jyn · 2025-11-02

**Soundness:** 2
**Presentation:** 3
**Contribution:** 2
**Rating:** 2
**Confidence:** 3

**Summary:**

The authors introduce NUBench, a negation benchmark that goes beyond cue based negations. The benchmark is manually curated, consisting of 2 types of datasets: the sentence-negation pair dataset and the multiple choice dataset. It consists of both training and evaluation sets, with evaluation being done in a completeness setting (predicting log probs) as well as a multiple choice setting (which is needed for non open souce LLMs given lack of access to log probs).

The authors report results using zero shot, few shot (demonstration examples) and supervised fine tuning (using the training set specified in the benchmark) on both the completion based as well as multiple choice setting across a variety of open as well as non-open source models.

**Strengths:**

* Negation is an important area to focus on. LLMs need to be tested more on negation since its a complex linguistic phenomena. The topic of the paper is well chosen.
* The paper is well written, and the theoretical bases for certain decisions (like types of negations and categorization) is well explained. These writing is especially important since a lot of readers aren't native to linguistics.
* In general, the approaches to collect data through humans (and failure to do properly through LLMs) is explained decently.

**Weaknesses:**

* I am worried about the soundness of either the dataset or the techniques given the fact that fine tuning barely moves the needle and in some cases has a negative effect. Its good they didn't observe this in the completion based setting. Is this because its harder to overfit on logits v.s a more direct text evaluation? I think some perplexity curves of training v/s dev and more analysis is absolutely needed here.
* For this paper to be considered ICLR level, we need a very thorough analysis of the results, gains and weaknesses of various models.This needs to take center stage, otherwise its hard to legitimize this work to the community.

**Questions:**

* Why the authors didn't make the decision to propose an evaluation only benchmark? What is the rationale here?

---

> ### Author Response · Authors · 2025-11-26
>
> We thank the reviewer for their thoughtful feedback. We appreciate the acknowledgment of the importance of negation in LLMs and the theoretical grounding of our work. Below, we address your concerns regarding the soundness of our SFT results and the benchmark design.
>
> **Q1. Soundness Concern (dataset & SFT)**: Why does SFT provide minimal or even negative performance gains, particularly in the option-selection setting?
> A1. SFT actually improves negation capability. As shown in Figure 3 and Table 12, the zero-shot performance after SFT is consistently higher than the zero-shot baseline (without SFT) across almost all models.
> The drop in option-selection evaluation is due to format mismatch, not lack of understanding. Our SFT was conducted using the Alpaca format (Input: Original Sentence, Output: Negated Sentence), which trains the model on a text generation task. However, the option-selection evaluation requires the model to output a specific label (A, B, C, or D). While SFT improved the models' internal understanding of negation (evidenced by improved completion-based scores), the instruction-following ability to output a multiple-choice label is reduced, particularly for smaller models.
>
> **Q2. Analysis of SFT and Perplexity Curves**
> A2. We agree that analyzing the training dynamics is essential to rule out overfitting. We will ensure to include the training and validation (dev) perplexity curves in the revised version of the paper.
>
> **Q3. Rationale for Including a Training Set**
> A3. We wanted to investigate whether LLMs can explicitly learn the rigorous logical rules of standard negation, rather than relying solely on few-shot in-context learning. Also, by providing high-quality sentence-negation pairs, NUBench serves as a resource for researchers to improve models' reasoning capabilities, not just to measure them.

---

### Note · Authors · 2025-11-26

I have read and agree with the venue's withdrawal policy on behalf of myself and my co-authors.